



# The 2017 Split wildfire in Croatia: Evolution and the role of meteorological conditions

Ivana Čavlina Tomašević[1,2,3], Kevin K. W. Cheung[4], Višnjica Vučetić[2], Paul Fox-Hughes[5], Kristian Horvath[2], Maja Telišman Prtenjak[3], Paul J. Beggs[1], Barbara Malečić[3], Velimir Milić[2]

[1]School of Natural Sciences, Faculty of Science and Engineering, Macquarie University, Sydney, NSW 2109, Australia
[2]Croatian Meteorological and Hydrological Service, Ravnice 48, 10 000 Zagreb, Croatia
[3]Department of Geophysics, Faculty of Science, University of Zagreb, Horvatovac 95, 10 000 Zagreb, Croatia
[4]E[3]-Complexity Consultant, Eastwood, NSW 2122, Australia
[5]Bureau of Meteorology, Level 5, 111 Macquarie St, Hobart, TAS 7001, Australia

*Correspondence to*: Ivana Čavlina Tomašević (ivana.cavlina-tomasevic@hdr.mq.edu.au / ivana.tomasevic@cirus.dhz.hr)

**Abstract.** The Split wildfire in July 2017, which was one of the most severe wildfires in Croatian history of this World Heritage site, is the focus in this study. The Split fire is a good example of wildfire-urban interface, with unexpected fire behavior including rapid downslope spread to the coastal populated area. Thus, it is critical to clarify the meteorological conditions behind the fire event, those that have limited the effectiveness of firefighting operations and the rapid escalation and expansion of the fire zones within thirty hours.

First, the Split fire propagation was reconstructed using radio logs, interviews with firefighters and pilots involved in the intervention, eye-witness statements, digital photographs from fire detection cameras, media and firefighting monthly journal. Four phases of fire development have been identified. Then, weather observations and numerical simulations using an enhanced-resolution operational model are utilized to analyze the dynamics in each phase of the fire runs. The synoptic background of the event includes large surface pressure gradient between the Azores anticyclone accompanied by cold front and a cyclone over southeastern Balkan Peninsula. At the upper level, there was a deep shortwave trough extending from the Baltic Sea to the Adriatic Sea, which developed into a cut-off low.

Such synoptic conditions have resulted in the annual maximum of Fire Weather Index and the highest monthly severity rating for July in the period 1981-2020. Combined with topography, they also provoke locally the formation of the strong northeasterly bura wind along the Adriatic coast. During the fire event, wind gust of nearly 25 m s-1 occurred. Low level jet (LLJ) has also been formally identified during an extended period, with a peak prior to the fire event possessing wind speed of over 21 m s-1 at a height of 600-700 m. Analysis of the upper-level jet also reveals that there was a deep tropospheric bura, which has facilitated the subsidence of dry air from the upper troposphere. In the mid to lower level, gravity wave breaking and turbulence mixing (as in the hydraulic jump theory) in the downslope bura wind further enabled the rapid drying at the surface.

Low level jet and strong downslope wind such as the bura are known to be related to many severe wildfire events worldwide, besides the antecedent hot and dry weather conditions and fuel loads. As has been demonstrated in this study, numerical

guidance that indicates the spatial and temporal occurrence of low level jet is highly implicative to explain the Split fire

evolution from the ignition potential to its extinguishment stage. Thus, in addition to the conventional fire weather indexes, such products are able to improve fire weather behavior forecast and in general more effective decision-making in fire management.

**Keywords:** wildland-urban fire, wildfire, bura wind, low-level jet, hydraulic jump, cold front, dry air intrusion

## 1 Introduction

Croatia lies within one of the world's most fire prone areas, the Mediterranean Basin (Fig. 1a; e.g., San-Miguel-Ayanz et al., 2013; Rundel et al., 2018). In recent years, Mediterranean Europe has experienced a significant number of wildfires that have caused tremendous casualties in terms of human life (Lagouvardos et al., 2019), natural destruction (Pausas et al., 2008) and economic disruption (Moreira et al., 2011). Especially concerning are fires that burn in so-called wildland-urban interface

(WUI), such as many Mediterranean coastal regions, where small touristic towns merge with natural areas (Bento-Gançalves and Vieira, 2019). Coincidence of a wildfire and extreme weather conditions within WUI can contribute to catastrophe. For example, the severe blaze within WUI in Eastern Attica (Greece) in 2018 took the lives of 102 people in less than 3 h (Lagouvardos et al., 2019).

Similar wildfire tragedies have occurred in many other regions globally. The deadliest wildfire in this century in

Australia, Black Saturday in 2009, killed 173 people, the majority in the first 12h following ignition (BoM, 2009), while extremely high death toll and economic loss occurred in the United States (California) in the first 24 hours of the Camp Fire (Brown et al., 2020). Extreme wildfires are rare (accounting for only 1% of fire occurrences) but cause more than 90% of damage (Strauss et al., 1989). Once ignited, rapid wildfire progression causing huge damage and high mortality mostly occurs in a short time interval during the wildfires' active period (Wang, 2011). Whether wildfire exhibits extreme behavior and

becomes a major threat largely depends on prevailing weather conditions (e.g., Lydersen et al., 2014; Hernandez et al., 2015). Indeed, it has been found that, with available flammable vegetation, weather conditions explain the severe behavior and rate of spread of crown wildfires observed within Mediterranean ecosystems (Pyne et al., 1996; Ruffault et al., 2017).

The local state of the atmosphere in days prior to wildfire and during wildfire activity is determined by synoptic scale weather systems. Linking synoptic features to extreme wildfire behavior or fire danger, and localizing major wildfires in



relation to existing fronts, low and high pressure areas, has long been the subject of fire-weather research (e.g., Beals, 1914; McCarthy, 1923). The majority of such studies found the most threatening synoptic patterns to be the ones creating conditions of increasing wind, accompanied by unusually low relative humidity, with an antecedent period of warm and dry stable conditions. Thus, the trigger for extreme wildfire behavior is an abrupt transition from so-called 'blocking' or stationary and persistent anticyclonic patterns to a low pressure environment, accompanied by a sudden change in wind direction and increase

in wind speed, often with little or no precipitation. Indeed, many severe wildfire cases have been associated with the surface pressure pattern appearing as a border between two different air masses, known as summertime dry cold front. This synoptic scale phenomenon was confirmed in the aforementioned case of the catastrophic Black Saturday fires in Australia in 2009 (Cruz et al., 2012; Engel et al., 2013; Dowdy et al., 2017), as well as in numerous other large wildfires that have occurred either prior or following a dry cold front passage in southeast Australia (e.g., Bond et al., 1967; Mills, 2005a,b; Long, 2006;

Fromm et al., 2006; Reeder et al., 2015) and United States (Schroeder et al., 1964; Brotak, 1977; Brotak and Reifsnyder, 1977). Although less often, cold fronts do sweep the Adriatic coast in summer. Research on synoptic conditions that occurred during 11 large (> 500 ha) wildfires in Croatia in the period from 1985 to 2010 showed association to cold fronts (e.g., Vučetić, 1987; 1992; Vučetić et al., 2007; Tomašević, 2012).

Another critical fire weather pattern includes synoptically-forced downslope winds, resulting from interaction of

prevailing flow with the underlying topography. Strong downslope winds are often related to sudden escalations in local fire danger levels and with rapid wildfire spread (e.g., Kondo and Kuwagata, 1992; Conedera et al., 1996; Sharples et al., 2010). Strong gusts cause abrupt surface drying and warming on the lee side of mountains through adiabatic compression and turbulent mixing (Whiteman, 2000; Abatzoglou et al., 2020). Foehn, a variety of downslope wind, has been related to severe wildfire behavior in the lee of the Rocky Mountains in the United States and southern Canada (Brewer and Clements, 2019),

the southeast Australian Alps (Marsh, 1987; Sharples et al., 2010), and the Southern Alps in New Zealand (Pretorius et al., 2020). The characteristic downslope wind associated with wildfires along the eastern Adriatic coast is the *bura* wind (local name for bora wind). It is the northeasterly, gusty and dry but cold wind that blows perpendicular to the mountain barrier of the Dinarides and mostly from northeast (NE) along the coast (Grisogono and Belušić, 2009). *Bura* is more frequent in northern than in southern Adriatic (including Split area), but it can be similarly severe (e.g., Horvath et al., 2009). Although *bura* is





more frequent in winter (Vučetić, 1991), with gusts up to 69 ms⁻¹ (248.4 kmh⁻¹; Vučetić and Vučetić, 2013) on the lee side of

the coastal range in the form of *bura* jets (e.g., Grisogono and Belušić, 2009; Telišman Prtenjak et al., 2015; Belušić et al.,

2018), severe episodes may occur during summer as well (Telišman Prtenjak et al., 2010). If *bura* coincides with a wildfire, it

dominates its behavior, as has been confirmed in multiple events (Kozarić and Mokorić, 2012; Tomašević, 2012).

       In addition to cold front and *bura* wind, low level jet (LLJ; Bonner, 1968) is another less obvious mesoscale/microscale

meteorological feature that has been found to coincide with large wildfires along the Adriatic coast (e.g., Vučetić et al., 2007;

Tomašević, 2012). Regardless of its synoptic background, LLJ is associated with a very strong wind shear and turbulence in

the atmospheric boundary layer (Byram, 1954). Rapid changes in wind speed and direction consequently result in rapid changes

in the direction, rate of spread and intensity of wildfires (Sharples et al., 2012), especially in areas of complex terrain such as

the Adriatic coast. Research on fire weather in Croatia is rare, and despite a few existing studies, many questions remain open.

Previous studies have also had limited knowledge on fire behavior and progression and, therefore, could not correlate it to

certain meteorological conditions.

       In July 2017 a severe wildfire occurred on the outskirts of Croatia's second largest city, Split, situated on the coast of

the Adriatic Sea. Due to its proximity to an urban area, the wildfire quickly captured public attention and became one of the

country's most significant wildfires in terms of firefighting resources involved in the intervention, burnt forest and agricultural

land, and the threat to people, property and infrastructure. The 'wind-driven' wildfire burned within complex coastal

orography, consisting of a steep mountain range backing the coastline. In the first 30 hours from its ignition, the wildfire

exhibited unusual behavior and was at times unexpectedly active. It was characterized by rapid progression, widespread

flaming and spotting; it easily transitioned to a crown fire, burned overnight without slowing down and on multiple occasions

spread rapidly downhill towards the city. An extreme fire weather event like in this case calls for special attention and provides

an opportunity to investigate meteorological factors that can lead to such a destructive and life-threatening phenomenon.

Therefore, the aim of this study is to analyze atmospheric processes related to the major fire runs during the Split wildfire in

Croatia, in order to improve understanding of the most dangerous fire weather conditions that can occur along the Adriatic

coast, and contribute to both fire weather forecasting and more effective decision-making in fire management.



The meteorological context of the Split wildfire is investigated in the subsequent sections. A description of the Split
environment and an overview of the wildfire's aftermath are given in Section 2. Section 3 describes the data and methods,
Section 4 details observed and modeled atmospheric conditions prior to ignition and in the first 30 most significant hours of
the Split wildfire, Section 5 provides further discussion and summary.

## 2 Overview of the Split wildfire

The most fire-prone area in Croatia is the Adriatic Sea coastline (Fig. 1a), together with its surrounding hinterland and
islands, of which there are more than a thousand in the Croatian archipelago. High fire risk is pronounced during summer
months, from June to August, when long dry spells and intense heat favor fire ignition and spread through highly flammable
Mediterranean vegetation including pine forests and shrubs. The majority of wildfires are human-caused (Mamut, 2011), with
the average annual burnt area of ~18 400 ha in ~2500 wildfires in the period 2006-2016 (DUZS, 2018). The burnt-area figure
escalated in 2017 with the total of ~87 000 ha in more than 4100 wildfires along the Adriatic coast, marking the worst fire
season in Croatian history.

Split is a historic and touristic city, listed as a UNESCO World Heritage Site (Kapusta and Wiluś, 2017). Its wider
urban area counts up to 300 000 citizens, with more than 720 000 tourists visiting in 2017 (Ministry of Tourism, 2018), mostly
in July and August, when wildfires are most frequent. The Split wildfire occurred on the last night of the Ultra festival, which
attracted more than 150 000 visitors into the city that weekend alone. The city is situated on a peninsula surrounded by gulfs
to the west and mountain and hills in the east. The wildfire started 15 km southeast from the city, in the valley between hills
parallel to the Adriatic coast and orientated north-west to south-east (Fig. 2). Further inland lies the highest mountain Mosor
(1339 m a.s.l.) with foothills, towards the Adriatic Sea, Makirina (marked as C in Fig.2; 723 m a.s.l.), Sridivica (B; 420 m
a.s.l.) and Perun (A; 533 m a.s.l.). The peaks are between 2 and 8 km from the sea, making this highly urbanized coast very
narrow. This type of topography, consisting of the steep mountain range rising from the coastline, can significantly influence
air flows and create complex atmospheric dynamics in the area. The hinterland landscape is dominated by Mediterranean
Aleppo pine forests (*Pinus halepensis* Mill.), scrub and maquis intermixed with small agricultural fields within scattered
villages. The area is well-known to be prone to fires, but mostly with minor wildfire incidents each year. The last significant



conflagration near Split, similar to the one from 2017 in terms of area burnt and firefighting demand, was in 2001 (Tomašević, 2012). However, that wildfire took 4 days to make the same impact as the 2017 fire had in under 30 hours (Francetić, 2017). The 2017 wildfire was stopped only 4 km from the city center.

The 2017 wildfire lasted nine days, from 16 to 25 July, and burned 5122 ha (Jovanović and Župan, 2017), most of which within 30 hours of ignition. The total cost of the Split wildfire is estimated at $US 20.6 million. It burned three houses and damaged 46 others, burned 18 cars, 11 olive groves and two greenhouses (DUZS, 2018). The plume from the wildfire crossed the Adriatic Sea and reached the coast of Italy, and it was clearly visible from space (Fig. 1b). Ash was observed up to 25 km south of the conflagration. Within the city, smoke drastically lowered air quality. The cause of the wildfire was declared to be of unknown origin. Given the size and rapid rate of spread of the fire, which made multiple runs into densely populated areas, it was very fortunate that no lives were lost as a direct result of the wildfire. Due to the intense fire activity, unexpected fire escalations, and enormous demands on property protection, mostly without aircraft support and with limited water supplies, additional firefighting resources and personnel from other parts of Croatia had to join the intervention, including ones from the closest island, which is unprecedented in Croatian firefighting history. In total 168 vehicles, 796 firefighters, and more than 200 soldiers were deployed. To date, firefighters refer to the Split wildfire as the "Mother of all fires".

## 3 Data and methods

### 3.1 Wildfire reconstruction

In order to correlate atmospheric conditions with extreme fire behavior a detailed wildfire reconstruction is provided before the meteorological analysis. Digital time referenced photographs from official firefighting cameras situated at the Zahod tower (Fig. 2) on the southeast peak of hill Perun (594 m a.s.l.) provided information on time of ignition, propagation and characteristics of the fire front, but only on its eastern side. The wildfire progression was mostly reconstructed from 3208 radio logs and 1124 emergency calls obtained from the Split Firefighting Brigade (SFB). This information, together with witness statements and interviews with firefighters and pilots, provided an insight into fire characteristics (flame height, crowning, smoke and plume), spotting, weather conditions on ground and upper-air turbulence. Together with interviews, a large number of photographs was collected. All the information gathered was geo-referenced and used to approximately define fire

isochrones. The reconstruction of the fire propagation and fire isochrones were plotted onto the total burnt area isochrone provided by the SFB.

## 3.2 Observations

Surface weather conditions were analyzed using meteorological data from the Split-Marjan station (122 m a.s.l.), the closest station to the wildfire (approximately 16 km west of the ignition location and 4 km from the closest line of final fire perimeter, Fig. 2). The Split-Marjan station is situated on the city of Split peninsula and has been operated by the Croatian Meteorological and Hydrological Service (DHMZ) since 1926, with automatic measurements since 2003. The meteorological variables used for this study include 10-minute data of air temperature, relative humidity, mean sea level pressure, precipitation

amount, mean and maximum wind speed and direction, and solar radiation, all from July 2017. Through the study, times are indicated in universal coordinated time (UTC), which is central European summer time (CEST) - 2h. All measurements were recalculated accordingly.

Antecedent weather conditions were analyzed using climatological assessments available from DHMZ. Assessments include the comparison of monthly, seasonal and annual air temperature and precipitation with the climatological period 1961–

1990 (from: https://meteo.hr/klima.php?section=klima_pracenje¶m=ocjena).

## 3.3 Fire danger rating

The Canadian Forest Fire Danger Rating System (CFFDRS) (Van Wagner and Pickett, 1985; Stocks et al., 1989) has been implemented in Croatia since 1982 (Dimitrov, 1982) and is used to alert firefighting agencies. The final product of the CFFWIS (Canadian Forest Fire Weather Index) system is Fire Weather Index (FWI), which is a combination of six sub-indices.

Along with the FWI, this study will also focus on the ISI (Initial Spread Index), one of sub-indices which represents the rate of fire spread in m min$^{-1}$.

## 3.4 Synoptic charts

The data used to examine the synoptic environment prior to and during the Split wildfire included synoptic surface and upper-level    analysis    obtained    from    the    German    Meteorological    Service    (Deutsche    Wetterdienst,    DWD,



www1.wetter3.de/Archiv/). The products used included 850-hPa and 300-hPa wind and relative vorticity charts, 500-hPa

geopotential (gpdam), surface pressure and relative topography (RT, at 500 m and 1000 m).

### 3.5 Numerical model

Numerical simulations were performed using the operational limited area mesoscale numerical weather prediction

model ALADIN/HR (ALADIN International Team, 1997). Details on model setup and configuration can be found in Tudor

et al. (2013, 2015). For the purpose of this study, ALADIN/HR model was initialized at 00 UTC for each day of the Split

wildfire, from 16 to 25 July 2017, with the hourly output data. Simulation ran with two nested domains (in operational use in

DHMZ) in 4 km horizontal resolution (ALADIN-HR44) up to 72 hours forecast. The outer domain covers a 1900 km x 1700

km area, while inner domain is zoomed on the area covering 550 km x 550 km over Croatia. ALADIN model also provides

dynamical adaptation of wind fields (ALADIN-HRDA) with 2 km horizontal resolution which has in a number of cases

improved near surface wind representation in complex terrain such as the Adriatic Sea coastline (e.g., Hrastinski et al., 2015).

Dynamically downscaled surface wind fields with a grid spacing of 2 km for the purpose of this study covered an additional

sub-domain of 250 km x 250 km around Split.

Numerous validation and verification methods, both in operational and in research context, applied over the years

confirmed that ALADIN model also provides very good representation of the vertical state of the atmosphere (e.g., Horvath et

al., 2009; Ivančan-Picek et al., 2016; Stanešić et al., 2019). Vertical grid in products of 4 km grid spacing is stretched with 73

hybrid sigma-pressure levels with the lowest vertical level at approximately 10 m above ground level, while dynamic

adaptation products have 15 vertical levels (with 8 levels in the first 1000 m). Vertical profiles in this case are simulated for

Split location (43.525°N, 16.506°E), and included air pressure, air temperature, dew point temperature, wind speed and wind

direction.

Finer-scale atmospheric features were additionally examined by vertical cross sections of horizontal wind speed and

direction combined with air temperature, relative humidity, potential temperature and $z$-wind covering 300 km horizontally

and 5 km in height. The location of vertical cross sections can be seen in Fig. 1a.





**3.6 Low-level jet definition and spatial distribution**

Here we introduce a new ALADIN model product, a spatial distribution of LLJ. Vertical profiles were simulated for

each grid point at 4 km resolution and plotted over inner domain over Croatia for each hourly time step. LLJ at grid point was

defined according to one of four criteria (Bonner, 1968):

- a wind speed maximum between 10 and < 12 ms$^{-1}$ with a wind speed decrease aloft by 4 ms$^{-1}$ up to the 3 km

    height, noted as LLJ criterion 0;

- a wind speed maximum between 12 and < 16 ms$^{-1}$ with a wind speed decrease aloft by 6 ms$^{-1}$ up to the 3 km

height, noted as LLJ criterion 1;

- a wind speed maximum between 16 and < 20 ms$^{-1}$, with a wind speed decrease aloft by 8 ms$^{-1}$ up to the 3 km

    height, noted as LLJ criterion 2;

- a wind speed maximum ≥ 20 ms$^{-1}$ with a wind speed decrease aloft by 10 ms$^{-1}$ up to the 3 km height, noted

    as LLJ criterion 3.

The LLJ criterion 0 was additionally implemented since some of the previous studies indicated that ALADIN may

underestimate near-surface wind speed (e.g., Vučetić et al., 2007). To our knowledge, a spatial distribution of LLJ speed and

height has never been applied in fire weather research to date.

**4 Results**

**4.1 Wildfire reconstruction**

The Split wildfire was characterized by four very active fire runs in the first 30 hours from ignition (Fig. 2). Those four

periods of broad fire spread accompanied by erratic fire behaviour and air turbulence will be noted as SPLIT 1 through 4.

SPLIT 1 will refer to the first 11 hours of the wildfire, or a period from the late-night ignition to the morning hours the following

day when fire activity slightly eased. Within this period, firefighting aircraft could not join the intervention due to air

turbulence. The SPLIT 2 period will refer to early afternoon fire reactivation and further spread of the fire zone with mosaic

fire front. SPLIT 3 will refer to the late afternoon escalation in fire activity around all zones with the most significant downhill

fire run into the city. The fourth and final period SPLIT 4 will refer to the night time downhill fire run into the eastern suburbs





of the city. It should be noted that during defined periods wildfire was simultaneously progressing and remaining active while also reactivating at locations impacted beforehand.

### 4.1.1 Burn period Split 1: 22:38 UTC (16 July) – 10 UTC (17 July)

The wildfire was reported in the evening on 16 July 2017 at 22:38 UTC (00:38 CEST on 17 July), 15 km east from the city, on the south foothill of Makirina (C, Fig. 2). Within minutes surveillance cameras (Z, Fig. 2) detected very fast fire growth. Wildfire developed under a very strong and gusty NE *bura* wind, which pushed the fire in SW direction, into the valley. However, between strong *bura* gusts fire progressed northwards, burning uphill Makirina (C), threatening villages at higher altitudes and the astronomical observatory. Depending on available fuels, wildfire easily transitioned to crown fire.

From 05 UTC to 13 UTC on 17 July 2017, firefighting aircraft made multiple attempts to join the intervention, but were unable to approach the site due to severe turbulence. According to fire officials, at one period during the early morning fire activity slightly eased and wildfire could potentially have been controlled with air assistance at higher altitudes while ground troops focused their suppression efforts on keeping the fire away from villages at lower altitude.

### 4.1.2 Burn period Split 2: 10 – 15 UTC (17 July)

The significant shift in fire activity occurred around 10 UTC on 17 July. While still flanking along the hill Makirina, mostly towards the north-west towards the city of Split, a southern flank of the fire front reactivated and spread further into the valley (Fig. 3a). Multiple spot fires created a mosaic fire front. Photographs from the camera at Zahod location (noted as Z in Fig. 2) revealed fire smoke rising in different directions within the valley and surrounding hills during the early afternoon (Fig. 3b). At Makirina hill (C; Fig. 2) smoke was rising in SW direction, within the valley in NW direction and at foothill of

Perun (A; Fig. 2) vertically. Wildfire easily crossed lower hill Sridivica (B) and burned upslope the north side of hill Perun (A, Fig. 2). At some locations wildfire crossed the hill A and threatened to run downslope towards the sea (which happened in the late evening of the same day during the SPLIT 4 period). At this time was prevented by the firefighting aircraft which could join the intervention only at south side of hill A between 13 UTC and 14 UTC. After 14 UTC weaker turbulence enabled firefighting aircraft to approach the fire burning in the valley, but it had only minor impact on it. By 15 UTC, the NW flank of

the wildfire, which was progressing towards the city, had travelled 6 km, 13 hours after ignition.



### 4.1.3 Burn period Split 3: 15 – 21 UTC (17 July)

During this burning period fire activity escalated around all fire zones. The NW flank of the wildfire, which was by 15 UTC located 10.5 km from the city center, turned SW and started its downslope run towards the city from the nearby hills and mountain Mosor (Fig. 2). The fire burned into dense pine forest in the NE higher altitudes of the wider city area. This area also

contains a possible minefield, remaining from the war in 1990s, which meant fire burned into plenty of long unburned dry fuels. As the main fire front entered heavy fuel, smoke and ash lofted into the extensive convection plume (Fig. 4a). Also, a number of spot fires were reported ahead of the main front, some ignited up to 500 m by flying pinecones. It is striking that in the first 20 min of this burn period, wildfire crossed additional 2.8 km, which makes the average forward rate of fire spread for this period to be 35 m min$^{-1}$. According to a firefighter witness, six fire whirls were spotted in the northern city suburbs,

along the foothill of mountain Mosor. Due to the wildfire's high intensity, erratic behavior and fast spread, constraining the propagation of the main fire front was not possible. Active fire suppression could only be organized in defensible space around people's homes. The situation within the city in this period can be described as chaotic. Fire threatened, among others, gas stations, substations and the city's main landfill. Observed spread rates within the outskirt suburbs were estimated to be from 500 m to 1 km per hour. The propagation of this flank of the wildfire was constrained due to fuel discontinuity and massive

suppression efforts of firefighters, self-organized citizens and military. This flank of the wildfire was stopped only 4 km from the historical city center, and brought under control by 21 UTC. Overall, in less than 6 hours wildfire travelled additional 6.5 km. Although wildfire did not travel far east, along the valley where it started, drastic reactivation of the fire front on this side occurred simultaneously with the downslope fire run into the Split urban area (Fig. 2), which contributed to chaos in already strained fire management. Firefighters on this side reported 3 km long fire front, extensive spotting and at one point flames up

to 30 m high.

### 4.1.4 Burn period Split 4: 21 UTC (17 July) – 04 UTC (18 July)

By this time the wildfire drastically reactivated on the hill Perun (A; Fig. 2). Wildfire crossed the hill multiple times on 17 July, but only around 21 UTC its activity escalated and could not be stopped before it ran downslope towards the sea. Wildfire burned into a native downy oak (*Quercus pubescens*) forest on the top of the hill and spread rapidly downhill reaching

narrow and densely populated coastal area at the bottom of the hill within minutes (Figs. 4b, d). Crown fire propagated down





slopes inclined at approximately 20°, and in less than 30 minutes burned 1 km of forest before it reached houses. This flank of the fire front was 700-800 m long, with the average forward rate of fire spread of 2 km h$^{-1}$ or 33 m min$^{-1}$. According to witnesses, pinecones from the burning forest on the hill started several isolated spot fires up to 800 m ahead of the fire front. Flames from the crown fire reached heights in a range from 10 to 30 m above the canopy. This flank of the wildfire was controlled around 4 UTC in the morning on 18 July 2017. The majority of 5122 ha burned by this time. Only small additional areas burned until the wildfire was declared contained nine days after ignition, on 25 July.

## 4.2 Antecedent conditions and fire danger rating

The summer season in Croatia in 2017 was extremely warm and dry with air temperature at the Split-Marjan meteorological station 3.1°C above average, and with only 6% of the 30-year (1961–1990) mean rainfall. Extreme weather conditions during the summer were extension of a long dry period that started in the preceding spring season. Spring was very warm and dry, with the last significant rainfall in Split two months prior to the wildfire (on 26 May, 10.5 mm).

The lack of precipitation accompanied by higher-than-average air temperature in the months prior to the wildfire led to continued drying of fuels in the region and consequently had an impact on fire danger rating. Fire danger was very high for more than 20 consecutive days prior to the Split wildfire. On the day of the fire, FWI reached its annual maximum and ISI reached the seasonal maximum (Fig. 5). This confirms that the most severe fire weather conditions in 2017 occurred exactly on 16 July, the first day of the Split wildfire.

Additionally, according to the definition of ISI, if it is greater than 18, then the estimated speed of a fire front is 18.3 m min$^{-1}$. Seasonal peak value of ISI (27.4) also pointed out that, along with rapid spread, wildfire may create multiple fire fronts and develop into a crown wildfire, the most dangerous type of fire. According to wildfire reconstruction, this type of fire behavior occurred exactly in the first 30 hours of the Split wildfire.

## 4.3 Surface synoptic conditions

The synoptic analysis revealed that prior to and during the first 30 hours of the Split wildfire there was a strong pressure gradient over the Adriatic coast (Fig. 6a). On 16 July and most of the day on 17 July Croatian territory was placed between front of the Azores anticyclone and rear of the cyclone over SE Balkan Peninsula. Consequently, the strong pressure gradient



over 600 km long coastline was created, with pressure varying from approximately 1023 hPa to 1010 hPa, which was followed

by an advection of strong NE airflow. This gradient remained strong in the morning on 17 July when aircraft reported severe

turbulence. The pressure gradient along the Adriatic eventually weakened on 18 July and was replaced by almost non-gradient

conditions which lasted for several days until a low pressure system on 24 July brought light rain over the fireground. These

conditions helped firefighters to completely extinguish the wildfire on 25 July 2017.

Model data corroborate the surface pressure analysis and depict the strong pressure gradient over the wildfire's area

prior to ignition and until midday on 17 July (transition from SPLIT 1 to SPLIT 2 period). The wildfire location (43.5°N,

16.6°E) of ignition was placed in the narrow band of tight pressure gradient between 1020 hPa to 1012 hPa over 100 km of N-

S line (between 43°N and 44°N; Fig. 9a). This tight pressure slightly eased during the day on 17 July (SPLIT 2 to SPLIT 4),

and was replaced by a non-gradient field in the midday on 18 July.

## 4.4 Upper-level trough and cyclone

The upper-level charts revealed that synoptic conditions coinciding with the Split wildfire featured a large amplitude

upper-level trough extending from the Baltic Sea in the north to the Adriatic Sea in the south (Fig. 6b). The trough amplified

in the hours prior to the wildfire. Around the time of ignition, the trough attained maximum strength and traveled slightly east,

placing the wildfire's area exactly on its west side. Analysis of 500 hPa chart (not shown) revealed stronger wind speed here

(25.7 ms$^{-1}$), accompanied by a 300 hPa jet stream (Fig. 6c; up to 46.3 ms$^{-1}$). This western flank of the jet stream and trough is

associated with air subsidence, which can be further confirmed by the advection of the vorticity maximum away from the

wildfire's location. The region right behind the vorticity maximum is linked to the strong sinking motion.

A large amplitude and shortwave trough are known to be dynamically unstable and also associated with fast upper-

level cut-off processes (Jurčec, 1989). The cut-off process in this case started at 00 UTC (Fig. 6b) and further deepened by 06

UTC on 17 July becoming a cut-off cyclone, which can be seen over SE Balkans and Greece. Upper-level trough acted as a

boundary between two airflows. On its west side, immediately above the wildfire's location, it brought a cool change with

strong NNE airflow, while on the east side it brought ESE airflow with cloudiness and development of storm centers, which

can be seen over SE Balkan Peninsula on satellite in Fig. 1b.



The model provides an accurate location of the upper-level shortwave trough stretched over the study area at 500 hPa,
at the time of the wildfire's ignition (Fig. 7a-c), however, the cut-off process appeared earlier (by 16 UTC on 16 July) in
ALADIN simulations in relation to synoptic analysis and a little dislocated towards the Adriatic Sea. By the time of the ignition,
Split wildfire was exactly on the western or rear edge of the upper-level cyclone, which caused the cool air outbreak from the
north of the continent (Fig. 7a), bringing very dry air (Fig. 7b) and leaving clear skies over the entire Croatian territory, as can
be seen in the satellite imagery (Fig. 1b). After the ignition, upper-level cyclone progressively dissipated until the midday on
18 July 2017. During the whole study period (SPLIT 1 to SPLIT 4) the Croatian territory was placed in a narrow dry area of
subsiding flow (Fig. 7b).

Wind pattern at 500 hPa also confirms the cool air outbreak from the north (Fig. 7c). A jet-like shape following a jet
streak and jet stream aloft embedded the NE circulation in the morning on 16 July. As jet streak was situated on the west side
of the trough, it pointed to its amplification, which occurred hours prior to the ignition. The band of accelerated air further
intensified and positioned the edge of its core immediately above the ignition location at ignition time.

## 4.5 Surface conditions and bura wind

Automatic measurements from Split-Marjan station recorded the cool outbreak as a drop in maximum daily air
temperature by 5°C, from 33.3°C to 27.0°C between 15 and 16 July. This was followed by a drop in relative humidity, which
remained between 18% and 38% for two consecutive days, on 16 and 17 July.

Simulated air temperature and relative humidity follow the in-situ observation data and give insight into broader
conditions in the mountainous outback where wildfire started. Maximum values here on 16 and 17 July were between 25°C
and 29°C (Fig. 9c and e), with minimum values on the night of the ignition between 13°C and 19°C, depending on the elevation.
The overnight relative humidity, during the first hours of wildfire, reached the maximum of 60% at the elevated terrain (Fig.
9b). Early morning on 17 July brought a drop in relative humidity as expected (Fig. 9d), however, relative humidity in the area
remained below 40% the following the night (between SPLIT 3 and SPLIT 4 period; Fig. 9f).

Wind measurements at Split-Marjan station confirm the NE airflow during the first 30 hours of the wildfire (Figs. 8b,
c). A sudden increase in wind speed is evident in the afternoon on 15 July, with the strongest gust of the month: 19.9 ms$^{-1}$.
Wind gusts remained strong throughout 16 July, although decreasing to 4.5 ms$^{-1}$ by the time of the wildfire's ignition (Fig.



8b). Wind speed and gusts increased again (to 12.7 ms$^{-1}$) in the morning on 17 July, at the time of the reported air turbulence

by firefighting aircraft. Wind speed slightly eased at times during the mid-day on 17 July, and intensified again right at the

time of a downslope run towards the city of Split (the SPLIT 3 period). Wind direction remained persistent as NE *bura* wind,

which can be also seen by direction of the fire smoke which was perpendicular to the coast and traveled across the Adriatic

Sea towards Italy (Fig. 1b and 4a). The smoke also caused a drop in the total solar radiation at Split-Marjan station (not shown).

Wind dropped in speed and changed direction to SW in the morning on 18 July, which helped firefighters to control the fire

spread. Light rain on 24 July (1.2 mm) and 25 July (1.6 mm), also the most significant rainfall in two months, additionally

helped to finally extinguish the wildfire.

The dynamical adaptation of ALADIN model at 2 km horizontal resolution gave more detailed spatial structure of

near-surface winds in the area. Model data reveals that during the SPLIT 1 period, *bura* wind in the coastal outback where the

wildfire was burning at the time (foothill of C; Fig. 2) had a speed between 5.5 ms$^{-1}$ and 8.0 ms$^{-1}$ with gusts between 13.9 ms$^{-1}$

and 24.5 ms$^{-1}$. *Bura* retained this strength by 5 UTC on 17 July, when the aircraft tried to approach the fire site (Fig. 11a). At

the same time the wind dropped in speed away from the coast. The area of a low wind offshore and perpendicular to mountain

range during *bura* flow is known as wake (Grubišić, 2004). This low wind zone corresponds to the successful aircraft operation

at another wildfire site on the island 35 km south, which burned simultaneously with the Split wildfire. It is worth mentioning

that the Croatian firefighting aviation is one of the rare operations which descend to 20 m or even 10 m height (Žugaj, personal

communication).

During the SPLIT 2 period, *bura* retained strength in the area closest to mountain Mosor, however, narrow bands of

weak wind started to appear over the continental area in the NE section of the domain (Fig. 11b). One such band of weak wind

was located over the Split peninsula, Perun hill (A, Fig. 2) and the outback valley where wildfire reactivated and started its

reverse spread. Weaker wind speed along the hill A also contributed to successful aircraft operation on its southern side. During

the SPLIT 2 period, wind was westerly along the southern foothill of A and northeasterly along the hill C (Fig. 2).

At the time of the SPLIT 3 downslope fire run, *bura* wind over the landward part of the city's peninsula, at the location

of the NW flank of the wildfire (Fig. 2), remained strong with speed between 5.5 ms$^{-1}$ and 10.8 ms$^{-1}$ and gusts to between 10.8

ms$^{-1}$ and 24.5 ms$^{-1}$ according to the model (Fig. 11c). The speed of *bura* and its gusts persisted during the most critical hours





of fire burning within the city, after which it eased down to between 3.4 ms⁻¹ and 8.0 ms⁻¹ with gusts between 8.0 ms⁻¹ and

13.9 ms⁻¹ until the evening on 17 July (end of SPLIT 3 period, Fig. 11d).

Although weakening in the broader Split area and in contrast to the previous 48 hours, the *bura* wind continued into

the late evening and during the SPLIT 4 period, preserving its aforementioned wind speed and gusts until morning on 18 July,

after which it further weakened and wind turned westerly.

### 4.6 Hydraulic jump and dry air subsidence

Vertical cross sections reveal a hydraulic jump-like structure over the coastal mountain slopes at the time of the

wildfire's ignition (Fig. 12a and 12c). The *bura* flow was strongest between 600 m and 1700 m above ground level,

immediately upstream of the wildfire's location, with the maximum horizontal wind speed close to 30 ms⁻¹. Above this strong

*bura* flow was a layer of weak NE wind at altitude between 2300 m and 4300 m. This deep layer of weak wind on top of the

wind maximum in the lee of the coastal range indicates a possible wave breaking below, which is the mechanism of a hydraulic-

like flow. The presence of the hydraulic jump was also suggested by the positive vertical wind component at the downstream

end of the hydraulic jump, with the maximum value of +2 ms⁻¹ at this side (in combination with -2.5 ms⁻¹ within the downstream

flow; Fig. 12b). Hydraulic jump flow culminated right at the time of the ignition, after which it dissipated by the end of the

SPLIT 1 period.

The acceleration of the *bura* flow within the 1 km height throughout the day on 16 July is also apparent from the

potential temperature in the same cross section line (Fig. 12c). While the potential temperature field did not change

significantly on the windward side of the *bura* flow indicating the statically stable lower atmosphere during the observed

period, on the left side of the panel, or above the Dinarides and Split area, isentropes deformed during the day of the wildfire

suggesting a decrease in stability here. By 23 UTC on 16 July (ignition time) isentropes became densely packed with steep

downward, nearly vertical slope right above the mountain crest in the vicinity of the wildfire and jump-like recovery

downwind, also indicating a hydraulic jump. Deformation of isentropes occupied a deep layer from 800 m to 3500 m height.

Together with accompanied hydraulic jump this dense packing of isentropes signal existence of the orographic gravity-wave

breaking, known to generate strong *bura* flows (Gohm and Mayr, 2005). The sharp potential temperature gradient shows the

gravity wave right above the leeward side of the Dinarides (Fig. 12c). Peak gravity-wave activity occurred at the ignition time,





after which it weakened until the following morning (end of SPLIT 1). A descending slope of isentropes above coastal
mountains at the ignition time, when the gravity-wave was the most amplified, clearly indicates the strong flow acceleration
and formation of the jet in the lee of the mountains.

Appearance of sharper potential temperature gradient was accompanied by a significant drop in relative humidity.
Cross section of relative humidity reveals that the most prominent dry air descent occurred right at the ignition time (Fig. 12c).
A tongue of low relative humidity ($< 30\%$) extended downward to 1300 m height coinciding with the most intense sloping of
isentropes. Moving forward in time, the model indicated the relative humidity drop for the entire vertical column above the
wildfire area, which from early afternoon on 17 July had relative humidity under 30%. This low relative humidity persisted
during the overnight hours between 17 and 18 July (SPLIT 3 and SPLIT 4 periods) in the first 1000 m height and decreased
further under 10% to 3500 m height above the wildfire. This dry air subsidence is in agreement with the upper-level analysis,
which also suggested a possible dry air subsidence due to position of the upper-level cyclone in relation to the wildfire.

**4.7 Low-level jet**

*Bura* flow meets the characteristics of LLJ (defined in Section 3.4). Mechanisms recognized to cause LLJ include
synoptic pressure gradients, cold front passage, mountain waves, cyclogenesis in mid-latitudes and upper-level jet streak
dynamics (Uccellini, 1980; Jurčec, 1992). Pseudotemps or vertical profiles from ALADIN model permit analysis of the
temporal evolution of LLJ during the *bura* flow since radiosonde measurements at the closest station (Zadar airport) were not
obtained during the study period. A sequence of vertical profiles was simulated for locations closest to wildfire at key times –
at village Srinjine in the coastal hinterland (relevant for periods SPLIT 1 and SPLIT 2) and Split-Marjan station (relevant for
periods SPLIT 3 and SPLIT 4; Fig. 13).

The vertical wind profiles reveal the existence of a strong LLJ with a peak wind maximum above the wildfire's
location two and half hours before the ignition (19.1 ms$^{-1}$, LLJ criterion 2). During the first few hours of the wildfire, LLJ
eased to criterion 1 and remained this strength until the end of SPLIT 1 period (Fig. 13a) and throughout the SPLIT 2 period
(Fig. 13b). At 12 UTC on 17 July the LLJ speed was 13.0 ms$^{-1}$ between 786 m and 891 m height. As hill C has 723 m elevation,
this corresponds to the plume direction at the top of this hill (upper right corner in Fig. 3b). The LLJ was not found in this area
after the SPLIT 2 period.



In general, LLJ appearance and temporal evolution in the rough topography of Mosor mountain and the one from the

coastal location of Split followed the same pattern throughout the study period. However, at the location of Srinjine, LLJ was

slightly weaker with higher positioned maximum. This discrepancy in height of a LLJ core between coastal and outback

location is in agreement with previous studies on *bura* flow that found the center of the maximum flow higher in the outback

and lower along the coast (Lepri et al., 2015). At all times at both locations, whether during the mature stage of the LLJ or in

its complete absence during the periods SPLIT 3 and SPLIT 4 (Fig. 13c and d), wind was persistently NE up to 3000 m height.

The vertical profile up to 10 km revealed that upper jet stream had a peak strength right prior to ignition (57.3 ms$^{-1}$)

and during the SPLIT 1 period. At all times during the study period, wind direction throughout the troposphere was N to NE

(with some exceptions in the first 1000 m height during the SPLIT 3 and SPLIT 4 periods), illustrating that this was a deep

*bura* event (Gohm and Mayr, 2005).

Vertical profiles of air temperature (Fig. 14c) reveal the absence of the inversion in both lower and upper troposphere.

The lower troposphere lacked the inversion at all times significant to strong *bura* flow, even during the hydraulic jump

appearance. Vertical profiles of both air and dew point temperature (Fig. 14c, d) reveal their considerably different values for

the entire study period, which indicates very dry conditions. Dry conditions might be explained by the complete absence of

the tropopause, which potentially led to larger vertical motion and dry air subsidence from the upper levels of the troposphere.

Dry upper-tropospheric air advection to the mid and lower troposphere was generated by the jet stream dynamics situated

above the study region. Vertical cross sections revealed that the dry air started to persistently dominate the fire ground after

the SPLIT 1 period until the end of the study period. The presence of a ribbon of dry air (Fig. 7b) with large potential vorticity

(Fig. 6c) suggests the translation and descent of a tropopause fold into the study area.

Previous studies suggested that LLJ is a weather phenomenon of considerable spatial extent (e.g., Vučetić, 1988;

Gohm and Mayr, 2005), however, this conclusion was drawn from simulated vertical profiles at various locations. This is the

first study to present the spatial distribution of LLJ in Croatia or, according to authors knowledge, elsewhere.

The spatial extent of LLJ reveals that the strongest jet defined as criterion 3 occurred over the highest coastal

mountains, extending from NW to SE, with parts of the flow stretching more than 100 km over the Adriatic Sea accompanied

by wakes in between. This spatial distribution of the strongest flow and wakes between the jet wind region over Adriatic





confirms the expected flow pattern formed by topographic incisions along the coast. The greatest extent of LLJ defined by
criterion 3 appeared 23 h prior to the wildfire (Fig. 15a) after which it slightly reduced its strength and coverage during the
midday on 16 July before it intensified again over the entire coast in the late afternoon hours, culminating two and half hours
before the ignition. The location of the wildfire during the SPLIT 1 period was situated in the wake-like region of the much
stronger flow in the outback, where the LLJ defined as criterion 3 coincided with the location of the hydraulic jump that
appeared in vertical cross sections. The LLJ at the wildfire's location during the morning on 17 July (end of SPLIT 1 period)
was classified as criterion 1 and 2 and stretched over the valley between hills A and C, right at the time of the reported
turbulence by firefighting aircraft. During the SPLIT 2 period the LLJ flow within the valley was classified by criterion 0 after
which it completely disappears from the area. Although LLJ appeared at the southeastern and northwestern edge of the wildfire
during the SPLIT 3 period, it gradually disappeared over the entire Adriatic region by the end of the study period or SPLIT 4.

## 5 Discussion and conclusions

### 5.1 Summary

The Split wildfire in July 2017 was one of the most severe wildfires in Croatian history given the size, unexpected
fire behavior and rapid spread which included two downslope runs into the densely populated area of the second largest city
in the peak of the tourist season. This study sets to answer several questions on meteorological conditions preceding this
wildfire event as well as those related to the rapid fire spread in the first 30 hours of ignition, noted as fire propagation periods
SPLIT 1 to SPLIT 4, within which burnt most of the total 5122 ha.

In the months leading up to the Split wildfire a prolonged period of extremely warm and dry conditions caused
continuous drying of fuels in the area and an increase of the fire danger which culminated exactly on the day of the ignition.
The annual maximum of FWI on 16 July 2017 at Split-Marjan station highlights the state of fuels as very dry and flammable
with the possibility for rapid fire spread, multiple fire fronts and crown fire, all of which occurred during periods SPLIT 1 to
SPLIT 4. These fire weather conditions mirror the state across the rest of the Mediterranean region affected by abnormal
drought and heat waves during the particularly severe and record breaking fire season of 2017 (e.g. Turco et al., 2019; Sanchez-
Benítez et al., 2018).



The sequence of severe antecedent meteorological conditions, combined with the specific synoptic situation that occurred prior to the ignition, contributed to the acute fire weather in the Split area. The favorable fire weather synoptic pattern in this case included: 1) a strong surface pressure gradient caused by the presence of an Azores anticyclone stretching towards central Europe and low pressure area over the southeastern Balkans and 2) long amplitude and shortwave upper-level trough extending from the Baltic Sea to Ionian Sea with the accompanying upper-level cut-off cyclone over SE Balkans. The synchronization of the low surface pressure area with the upper tropospheric trough produced a deep northeasterly *bura* flow over the Adriatic Sea. Deep *bura* flow, in contrast to shallow *bura*, extends throughout the troposphere and is typical for colder months (Grisogono and Belušić, 2009). As aforementioned, *bura* is a gusty downslope windstorm that blows from NE quadrant perpendicular to Adriatic coast and the adjacent Dinarides. The general criteria for severe *bura* is mean hourly wind speed greater than 17 ms$^{-1}$ for at least one hour (Vučetić, 1991). A severe *bura* downslope windstorm prevents road traffic between inland and coastal parts of Croatia and poses a great danger to aircraft. In this case, *bura* coincided with the wildfire ignition and strongly contributed to it becoming a large conflagration. Although *bura* in this case was weaker (with mean wind speed up to 10.5 ms$^{-1}$ and gusts up to 19.9 ms$^{-1}$ at Split-Marjan station) and does not fulfil the criteria for severe *bura*, it occurred during summer when such episodes are rare. *Bura* dominated the fire ground during each of the most significant wildfire progression periods from SPLIT 1 to SPLIT 4.

Based on the nexus of meteorological and fuel conditions in combination with complex topography, the most significant fire progressions during the Split wildfire from July 2017 can be explained as follows:

1) Both synoptic and upper-level conditions that coincided with the wildfire ignition are recognized to be among the most dangerous in fire weather literature. Strong surface pressure gradient with a source of dry air from the upper atmosphere that was transported to the surface by the hydraulic *bura* flow led to rapid fire growth immediately following ignition. In the first few hours of the nighttime SPLIT 1 period, strong NE *bura* pushed the fire downhill on south facing slopes of hill C (Fig. 2), into the valley, where the fire was eventually stopped by firefighters. Wildfire also burned upslope on hill C for two reasons. The first is due to buoyancy effects on flames and smoke between *bura* gusts and the second is potentially due to eddies and rotors in the lee, under the accelerated LLJ stream embedded in the *bura* flow (Gohm et al., 2008). This is, however, yet to be confirmed by numerical simulations of higher resolution.





2) The complexity of the flow at the wildfire's location was especially pronounced during the SPLIT 2 period (Fig. 3b). The sudden fire reactivation and its run downhill of C (Fig. 3a) surprised firefighter crews who had to redefend settlements in the valley that had been considered safe from the fire burning at higher altitude. Why the fire front could return into the valley and burn upslope on hills B, and afterwards A, may again be explained by the vertical wind profile which revealed lowering of both LLJ speed and height. By lowering its height, the core of LLJ now coincided with the top of the hill C where the wildfire was burning. As wind dropped in speed it may have resulted in more laminar and attached flow over the terrain which therefore pushed the fire again downslope of hill C with flying embers creating a mosaic fire in the valley. The LLJ weakening is related to daytime *bura* weakening, typical for a *bura* episode in its decaying stage (Gohm and Mayr, 2005) as was the case during the SPLIT 2 period.

3) The total fire escalation around all zones occurred during relatively benign fire weather conditions. *Bura* weakened by the beginning of the SPLIT 3 period (Fig. 11c) and the firefighting aircraft could join the intervention. However, the location of the wildfire at the time together with local atmospheric conditions are likely to be crucial for the rapid downslope fire run into the city area. By the beginning of the SPLIT 3 period, the NW flank of the wildfire (Fig. 2) burned into abundant dry fuels on the city edge on the slopes of Mosor. Covered by dense pine forest and long unburnt fuels this elevated terrain was aligned with the *bura* flow. The NE *bura* was still moderate to strong in this elevated area, contributing to a channeling effect and pushing the fire down the SW oriented slopes, towards Split. Such dynamic fire channeling is considered impossible to control due to high fire spread rate and intensity (Sharples, 2009), as was the case in this event. Furthermore, the rugged terrain with favorable fire weather and plenty of dry fuel available caused fire whirls and spotting. Dynamic channeling can also trigger evolution of pyro-cumulonimbus, and although not confirmed in this case, a large plume generated by the fire during the SPLIT 3 period signaled highly active fire behavior. The extensive NE plume (Fig. 1a) was sheared off sharply at approximately 4500 m altitude (Fig. 4a), consistent with the strong NE wind at this height found in vertical profiles.

Simultaneously, mosaic fire that was still flanking in the higher elevated valley between hills A and C on the eastern side of the wildfire (Fig. 2) merged into a single fire front. Intensification of the wildfire on this side was most likely caused by burning into heavier fuels and turbulent effects associated with the LLJ that persisted in the surrounding mountainous area.





4) Another downslope fire run during the nighttime SPLIT 4 period can be explained by moderate *bura* in the area (Fig. 11d) which pushed the wildfire over the top of hill A towards its southern side (Fig. 2). Its downslope run was therefore amplified by *bura* and additionally favored by nighttime reduction in relative humidity, most likely caused by dry upper

tropospheric air drawn down to the surface by the daytime mixed layer during the previous fire progression period SPLIT 3. Furthermore, on its downslope path the fire burned into downy oak forest resulting in significant fire escalation before it reached the urban area in the foothill of A in a matter of minutes (Fig. 4c and 4d). Again, such fire behavior is extremely dangerous for fire fighters, communities and assets in the path of such a rapidly advancing downslope fire.

**5.2 Further discussion on the dynamics of bura and LLJ**

In general, *bura* flow over the Adriatic can be described by dynamic processes presented in hydraulic theory (Long, 1953) where orographic wave breaking plays a key role for strong surface downslope windstorm occurrence (e.g., Smith, 1985; Vučetić, 1993). The theory includes acceleration of the flow upslope as well as an abrupt acceleration of the flow downslope in the lee with a hydraulic jump gradually restoring subcritical conditions (Cesini et al., 2004). Hydraulic jump is a frequent feature of strong *bura* flow over the Adriatic (e.g., Grisogono and Belušić, 2009). Although there are numerous studies

investigating the bora in the Adriatic (especially in the northern part during the cold part of the year, e.g., Horvath et al., 2009; Grubišić, 2004; Šoljan et al., 2018), in this study, the Split wildfire case study presents opportunity to analyze summer time moderate *bura* case over mid-Adriatic and to go a step further to relate it to the reconstructed wildfire behavior.

The only study on deep *bura* flow in the mid-Adriatic region in winter time suggests that hydraulic theory can be applied here if an upstream *bura* layer is sufficiently deep (i.e., 5 km in contrast to usual *bura* depth of between 2 km to 3 km;

Jurčec and Visković, 1989). Vertical wind profiles up to 10 km in the case of Split wildfire showed that NE wind extended throughout the troposphere (Fig. 14). Similar to aforementioned winter deep *bura* case study, the Split event lacked temperature inversion, which is usually assumed to exist above the *bura* layer and according to which it is possible to determine the top of the disturbed flow. In the case of the Split wildfire, temperature inversion or critical layer were not simulated with the ALADIN model. However, mesoscale models are known to underestimate inversions or stable layers and, when compared to radiosonde

measurements, the ALADIN model was found to underestimate the inversion layer in previous wildfire analyses (Vučetić et al., 2007).





Hydraulic flow is found to coincide with the Split wildfire ignition. It is marked by a wave-breaking aloft, an abrupt tilt of streamlines, accelerated wind on the leeward slopes and strong turbulence immediately above (Sharples, 2009; Whiteman, 2000; Smith, 1985; Jurčec and Visković, 1989). This type of flow is found in the most destructive wildfire in

California history, the 2018 Camp Fire, where hydraulic jump structures were linked to erratic surface winds causing the lifting of firebrands during the wildfire (Brewer and Clements, 2019). Similar hydraulic flow indicated a downward transport of energy and momentum during the deadly 2018 wildfire in Attica Region in Greece (Kartsios et al., 2020).

### 5.3 Concluding remarks

LLJ is of interest here not only as a phenomenon itself, but because of its effect on a wildfire behavior and aircraft

operations. Previous studies suggest that LLJ is associated with turbulent kinetic energy that can be mixed down to the fireground and cause rapid fire growth (Charney et al., 2003). Early US research (Byram, 1954) described vertical profiles similar to those found in this case as the most dangerous for fire weather, especially in the mountainous area. The reason for that lies in the intersection of the LLJ core and the elevated forested terrain, which in the case of a fire ignition can lead to blow-up fire behavior. Also, the fire behavior characteristics described for this type of wind profile include possible appearance

of fire whirlwinds, which were observed during the SPLIT 3 period.

LLJ has been found to coincide with all wildfires larger than 500 ha along the Adriatic coast in the period 2001–2011 (Tomašević, 2012; Mifka and Vučetić, 2012). However, it is important to note that although in the majority of those cases LLJ appeared in *bura* driven wildfires, LLJ has also been generated in different types of synoptic forcing (e.g., Mifka and Vučetić, 2012).

Rapid intensification of wildfires associated with a LLJ is reported in international literature as well. LLJ generated in the upper-level frontal zone contributed to the turbulent downward mixing of high momentum into the Mack Lake wildfire in 1980 in the Great Lakes Region, USA, and most likely caused the rapid fire spread reported in that event (Zimet et al., 2007; Charney et al., 2003). The fastest fire growth in a single-day was recorded in the Rocky Mountains Canyon Creek fire after LLJ became the dominant atmospheric feature in the area (Sharples, 2009). The 'blow-up' fire day with the unusually severe

fire weather during the catastrophic fires of Ash Wednesday in 1983 in south-eastern Australia included a deep tropospheric trough which prefrontal winds were accompanied by LLJ (Mills, 2005a).





Another significant finding from the Split wildfire, documented in association with severe fire weather conditions found in other catastrophic wildfires, is the influence of dry air subsidence. Descent of dry air occurred in conjunction with upper-level trough and jet stream dynamics above the study area. The subsidence process started 24 hours prior to the wildfire, with

the dry air descending sharply towards the wildfire right at the ignition time (SPLIT 1 period). This dry air descent was enhanced by the topographically-induced hydraulic *bura* flow on the downstream side of Dinarides. The dry air was further transported towards the already fast-growing wildfire with the deepening of the daytime mixed layer on 17 July 2017 (SPLIT 2 and SPLIT 3). These processes resulted in significant reduction in relative humidity during the downslope fire runs during SPLIT 3 and SPLIT 4 periods.

Australian cases related dry air subsidence with an abrupt reduction in surface relative humidity and consequently to rapid drying of fuels and extreme wildfire behavior during the catastrophic fires in Canberra in 2003 (Mills, 2005b) and Eyre Peninsula in 2005 (Mills, 2008a). Meteorological analysis of these wildfires found the mechanisms linking a reservoir of upper dry air with the surface. These mechanisms include dry convective turbulence in daytime mixed layer, fronts and topographically-induced flow on the downstream side of mountainous barrier (Mills, 2008b). Regarding the cause for the

upper-level subsidence some authors found that the strong descent motion that coincided with the increased fire activity was associated with tropopause fold (e.g., Mills, 2008a). These results are in agreement with the findings in the Split wildfire event.

Analysis of the Split wildfire leads to better understanding of *bura* driven wildfires within the complex topography of the mid-Adriatic region in Croatia and, moreover, towards application of LLJ spatial and temporal distribution in the future. It has been confirmed that LLJ is related to the most destructive wildfires in the area. Therefore, the information on LLJ provided

from the ALADIN model has the potential to improve fire weather forecasts. As LLJ spatial distribution is available in the 72 hours forecast range, it is possible to detect these phenomena days in advance. However, prerequisites such as long-term dry and warm weather conditions and, consequently, high FWI are necessary. LLJ, as an operational model product can identify locations where weather conditions are favorable for erratic fire behavior, especially if it coincides with other synoptic features such as dry air subsidence and, additionally, with a range of extreme mesoscale mechanisms enabling the downward mixing

of dry air to the fire ground.
Some previous studies proposed development of a new generation of fire weather indices that would highlight areas of LLJ intersection with a deep daytime mixing layer (e.g., Charney et al., 2003; Mills, 2005a). This kind of index can be designed for the Croatian region in the future. An operational model-based LLJ product such as that presented here could provide a pathway in that direction and, meanwhile, serve as a complementary information to FWI risk estimate and forecast. However,

in-depth hindcast verification should be conducted beforehand, i.e., to estimate high FWI and LLJ appearance and predict subsequent wildfires potential. Temporal evolution of LLJ can, among other data, assist in prediction of fire behavior in ongoing wildfires. All meteorological indicators found in this case study are likely to significantly contribute to better understanding and estimation of fire risk than those derived only from fire danger indices.

Only an operational numerical weather prediction (NWP) model with limited outputs has been utilized here. An

additional set of NWP model simulations at finer resolution for this wildfire case will be conducted to investigate the smaller scale *bura* flow features and LLJ impacts to the Split wildfire characteristics in more detail. In recent decades extreme wildfires around the world have demonstrated their destructive power, creating even their own weather and producing dangerous phenomena such as fire whirls, tornadoes or fire storms generally. As the Split wildfire also demonstrated unprecedented fire behavior, it is very likely that the energy released from the wildfire influenced the surrounding atmosphere. To investigate this

matter, it is in addition planned to prepare coupled fire-atmosphere simulations for this case study.

The systematic analysis of extreme wildfire events, such as the Split wildfire here, is also useful to derive a series of recommendations or lessons learnt to support fuel reduction practices, increase awareness of potential extreme events and prevent their occurrence in the Mediterranean region and other similar areas globally. The results are expected to contribute to better prediction of fire activity by fire management agencies, resulting in improved planning processes and capability,

including estimation of future fire regimes and exposure as a key adaptation element.

**Data availability.** The data can be provided by the corresponding authors upon request.

**Author contributions.** IČT, KKWC and VV designed the study; IČT and VV performed the calculations; IČT and KH prepared the visualizations; IČT, KKWC, VV, PFH and MTP analysed the data; IČT prepared the manuscript draft; KKWC, VV, PFH, MTP, KH, PJB, BM and VM reviewed the results and edited the manuscript; KKWC, VV, PFH and MTP supervised

the research.



**Competing interests.** The authors declare that they have no conflict of interest.

**Acknowledgement.** The first author (IČT) is supported by the Macquarie University Cotutelle Scholarship between Croatia and Australia.

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

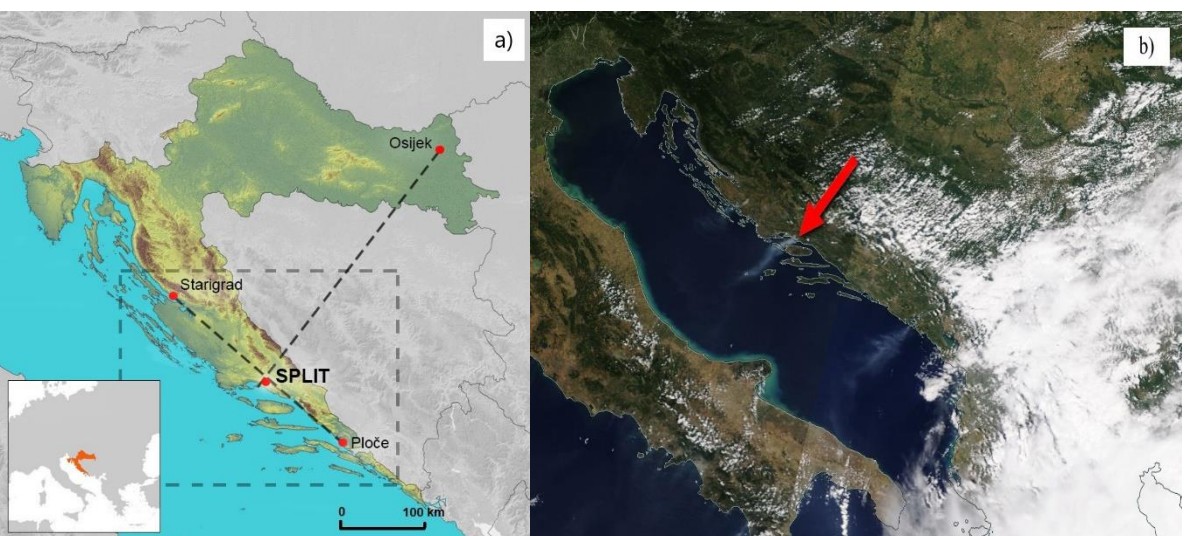

**Figure 1: (a) Location of the Split wildfire in Croatia with positions of vertical cross sections (dashed lines) and location of the inner nested domain used in the ALADIN model simulation (dashed rectangle) and (b) Terra satellite MODIS image on 17 July 2017 showing active fire areas along the Adriatic Sea coast (https://worldview.earthdata.nasa.gov).**


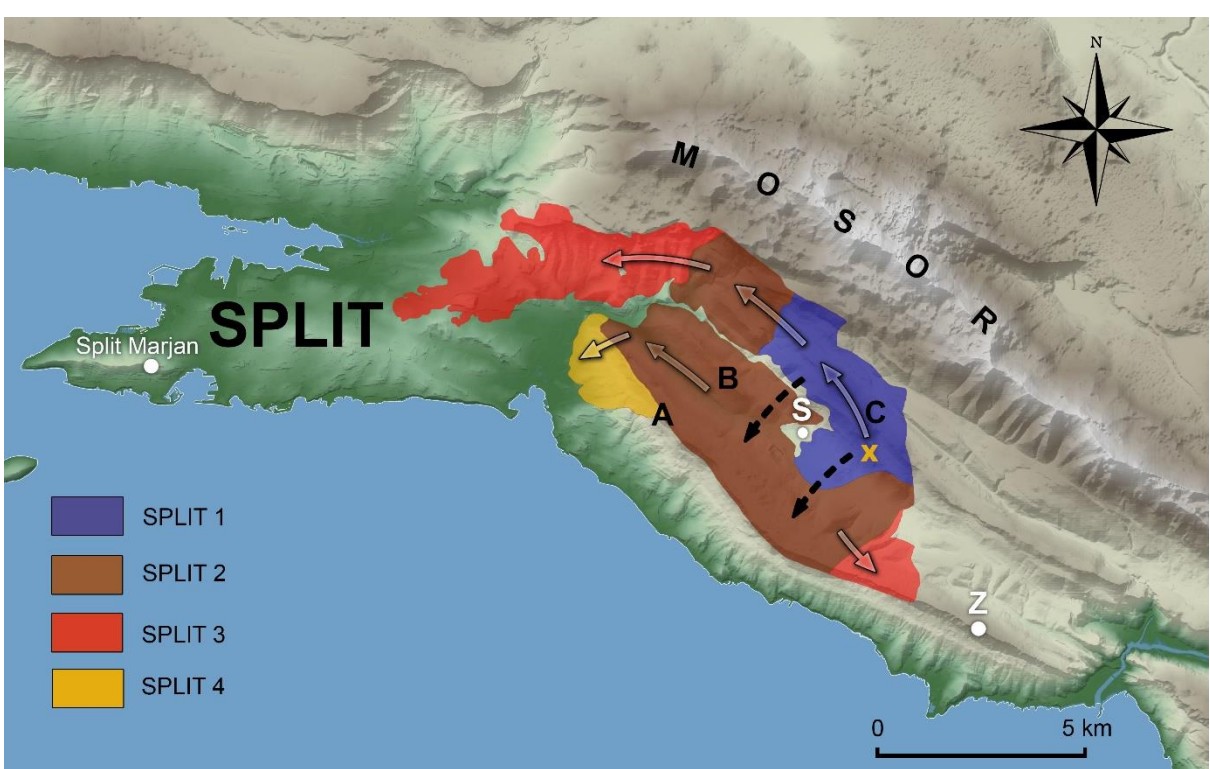

**Figure 2: Map of the Split wildfire with the final perimeter (according to initial information from the SFB) and four prominent progressions in growth, noted as SPLIT 1 to SPLIT 4, over the first 30 hours from ignition (from 22:38 UTC on 16 July 2017 to 04 UTC on 18 July 2017). Ignition location is noted as X. White dots indicate locations of Split-Marjan meteorological station, Zahod tower with cameras (noted as Z) used for fire detection and surveillance, and location of village Srinjine (noted as S). Split-Marjan and Srinjine are locations used for vertical profiles (pseudotemps). Letters indicate hills Perun (A; 533 m a.s.l.), Sridivica (B; 420 m a.s.l.) and Makirina (C; 723 m a.s.l.), all part of Mosor (1339 m a.s.l.) mountain range. (Basic topography from geoportal.dgu.hr).**





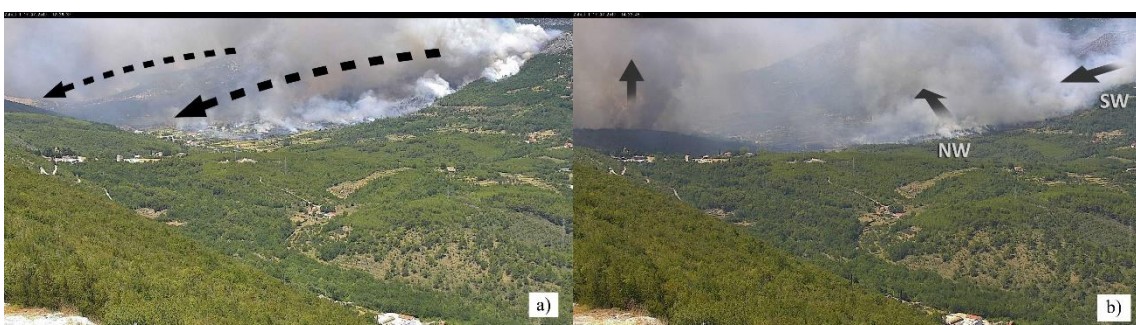


**Figure 3: Photographs from Zahod location of (a) mosaic fire front and wildfire spread down into the valley in SW direction between SPLIT 1 and SPLIT 2 period (at 10:39 UTC on 17 July) and (b) fire smoke rising in different directions in early afternoon hours during the SPLIT 2 period (at 12:30 UTC on 17 July).**


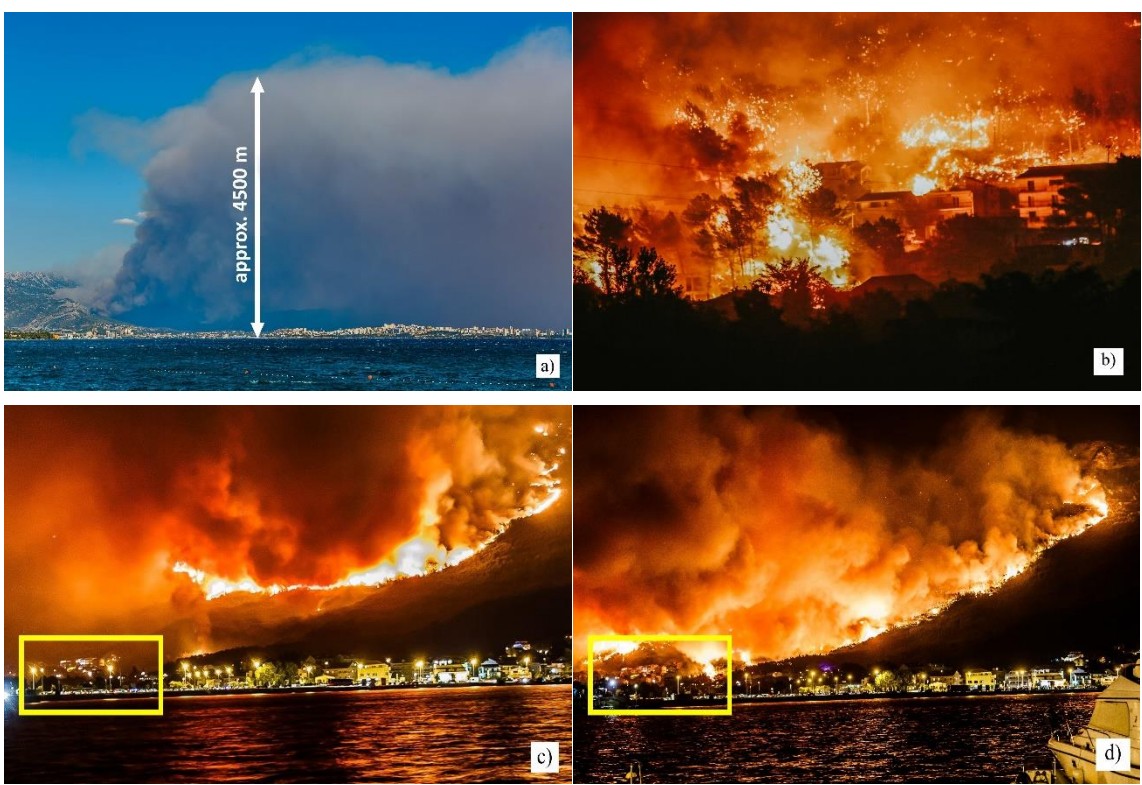

**Figure 4: (a) Fire smoke in afternoon hours (15:09 UTC) on 17 July during the SPLIT 3 period, (b) fire burning into the highly populated coastal area, (c) wildfire's downslope run into the coastal area on the south side of hill Perun (A) at 22:05 UTC and (d) 13 minutes later, at 22:18 UTC on 17 July, all during the SPLIT 4 period (photos a, c and d photographed by Zvonimir Barisin and b by Damira Kalajzic).**






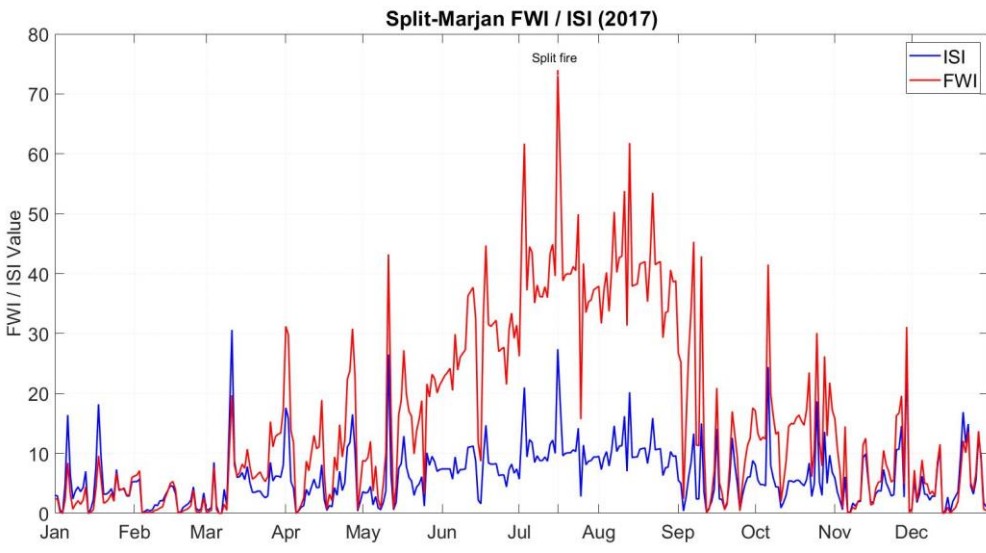

**Figure 5: Daily course of Initial Spread Index (ISI) and Fire Weather Index (FWI) at 12 UTC from 1 January 2017 to 31 December 2017 at Split-Marjan meteorological station.**




**Figure 6: Analysis charts for Europe at 00 UTC (approximately two hours after the ignition) on 17 July 2017 of (a) mean sea level pressure (hPa; black contours) and fronts, (b) 500 hPa geopotential (gpdam; black contours), surface pressure (hPa; white contours) and relative topography RT 500/1000 (gpdam; coloured) and (c) 300 hPa wind (kt, where 1 kt = 0.51 ms$^{-1}$; wind barbs) and relative vorticity (10$^{-5}$s$^{-1}$; coloured). Split wildfire location on charts is indicated as red dot. (The charts are available from: http://www1.wetter3.de/archiv/.)**


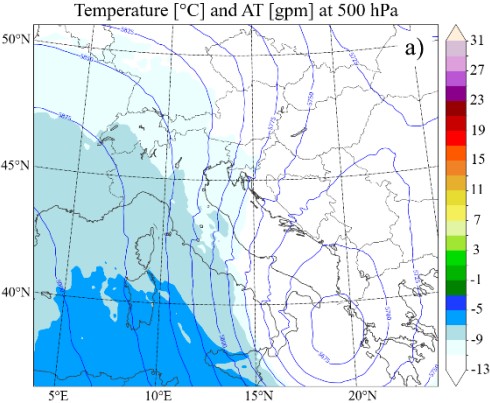

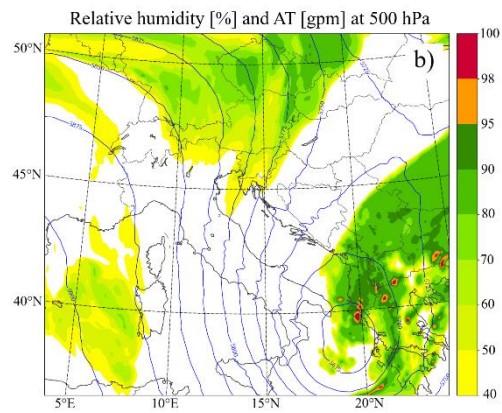

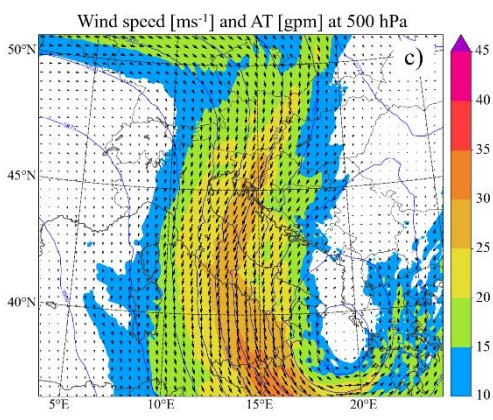

**Figure 7: (a) Temperature (°C; coloured), (b) relative humidity (%; coloured) and (c) wind speed (ms⁻¹; coloured), all including AT (gpm; blue contours) at 500 hPa from ALADIN-HR44 model valid for 16 July 2017 at 23 UTC.**


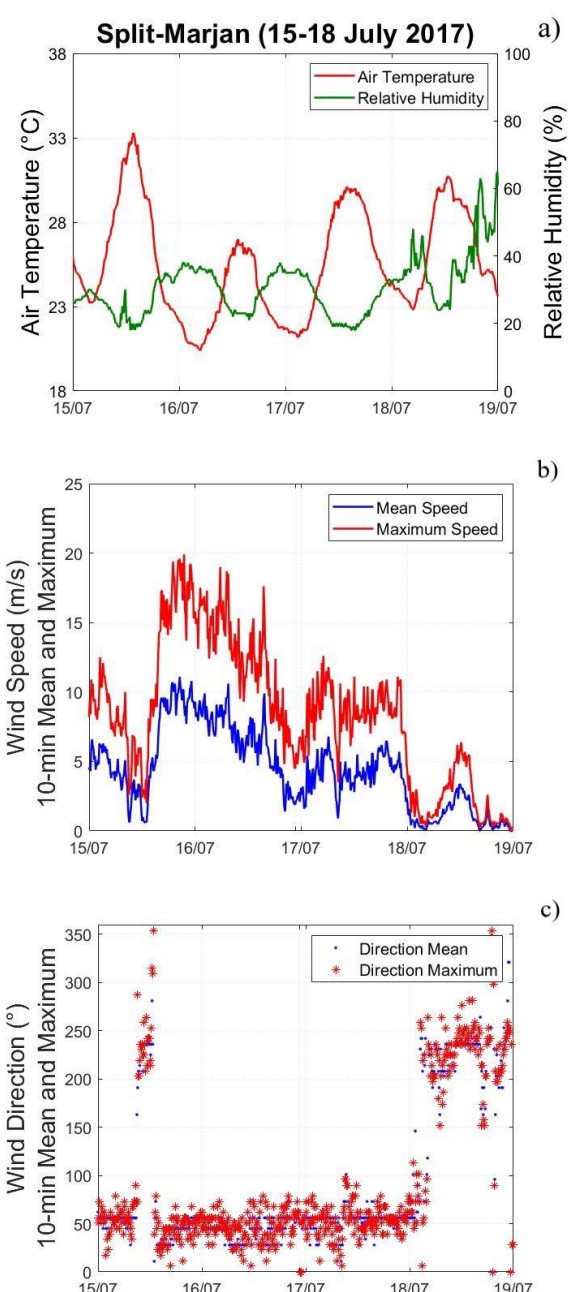

**Figure 8: Split-Marjan automatic weather station 10-minute observations of (a) air temperature (°C) and relative humidity (%), (b) mean and maximum wind speed (ms$^{-1}$) and (c) mean and maximum wind direction (°) from 15 to 18 July 2017.**



**Figure 9: (a) Mean sea level pressure (hPa; coloured), (b) relative humidity (%; coloured) at 2 m, both valid for 23 UTC on 16 July, (c) air temperature (°C; coloured) at 2 m valid for 13 UTC on 16 July, (d) relative humidity (%; coloured) at 2 m valid for 15 UTC, (e) air temperature (°C; coloured) at 2 m valid for 13 UTC and (f) relative humidity (%; coloured) at 2 m valid for 23 UTC, all valid for 17 July 2017 from ALADIN-HR44 model.**

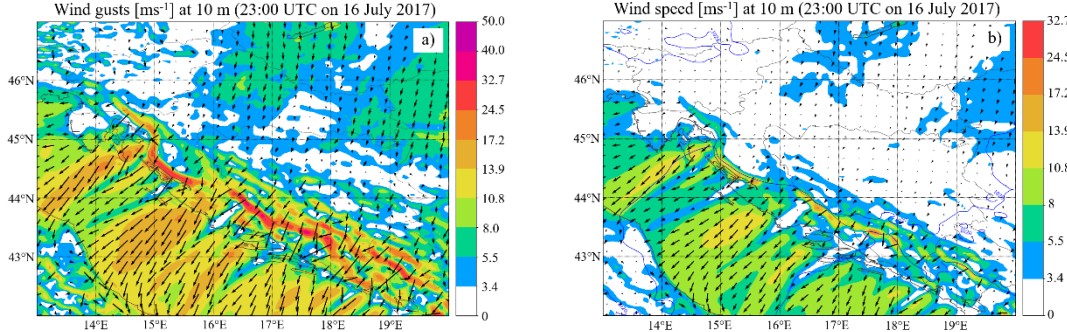

**Figure 10: (a) Wind gusts (ms⁻¹; coloured and array) at 10 m, (b) wind speed (ms⁻¹; coloured and array) at 10 m and MSLP (blue contours), both valid for 23 UTC on 16 July 2017 from ALADIN-HR44 model.**



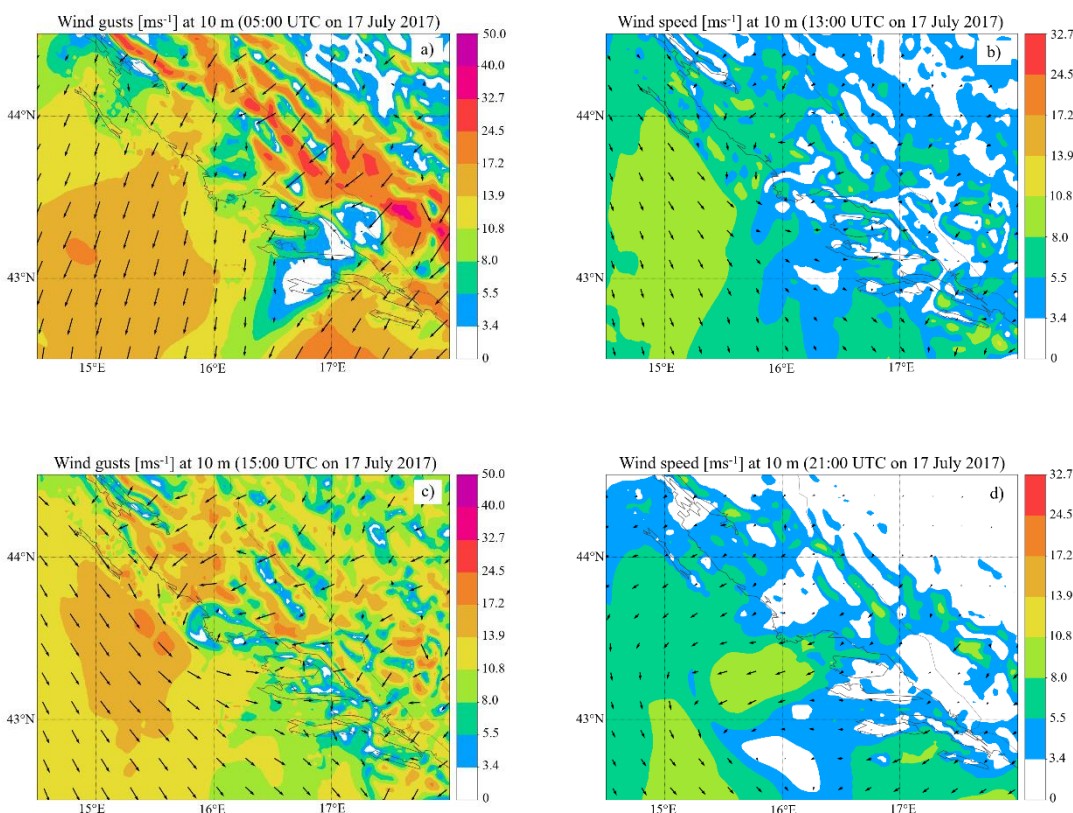

**890**  **Figure 11: (a) Wind gusts (ms⁻¹; coloured and array) at 05 UTC, (b) wind speed (ms⁻¹; coloured and array) at 13 UTC, (c) wind gusts (ms⁻¹; coloured and array) at 15 UTC and (d) wind speed (ms⁻¹; coloured and array) at 21 UTC, all valid for 17 July 2017 at 10 m from ALADIN-HRDA model.**



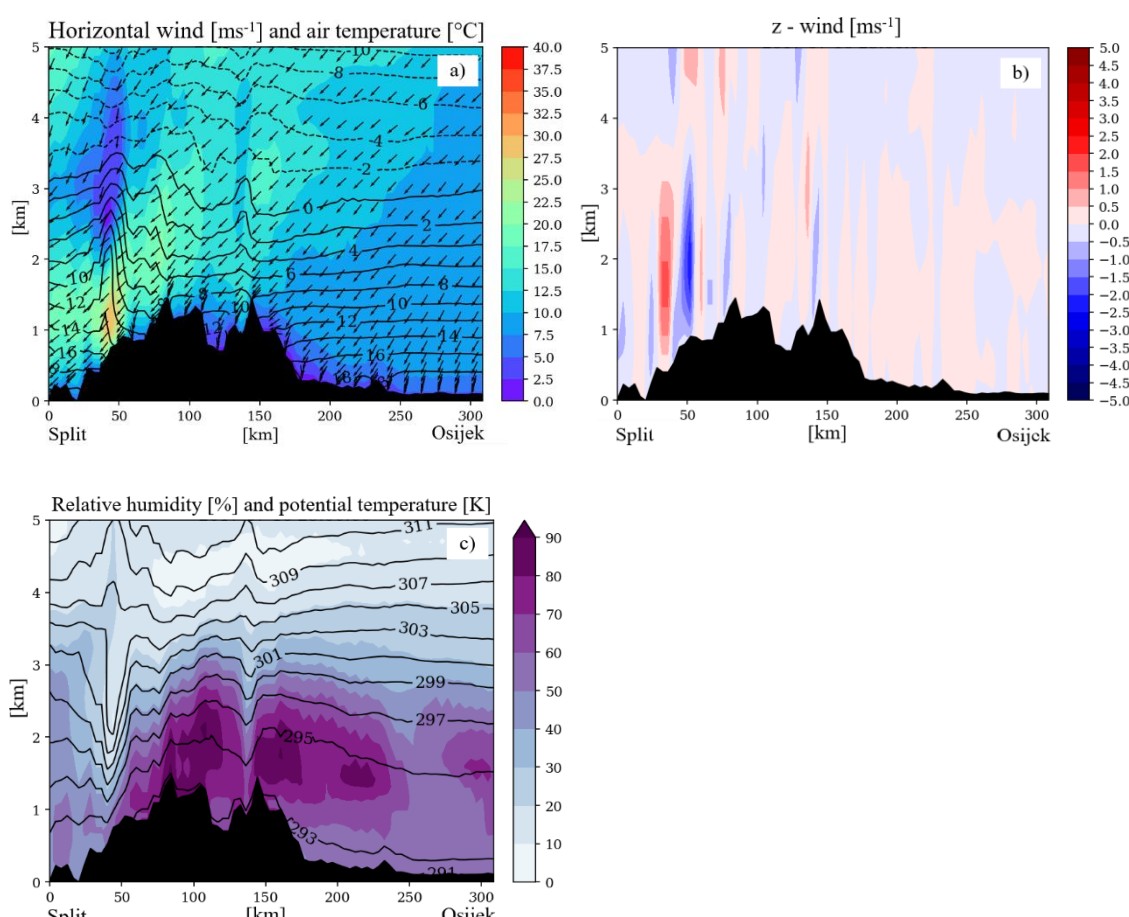


**Figure 12: Vertical cross sections from ALADIN-HR44 model of (a) horizontal wind speed (ms⁻¹; coloured) and direction (array) and temperature (°C; black contours for ≥0°C, dashed contours for < 0°C), (b) z-wind (ms⁻¹; coloured) and (c) relative humidity (%; coloured) and potential temperature (K; black contours every 2 K) all valid for 23 UTC on 16 July 2017 from ALADIN model. The bottom black area depicts the terrain. Location of cross section between cities Split and Osijek is indicated in Figure 1a. Each section is 300 km long and 5 km high, oriented northeast to southwest and perpendicular to Adriatic coast with Split situated approximately 20 km from the left bottom corner. Air flow in each panel is from right to left.**



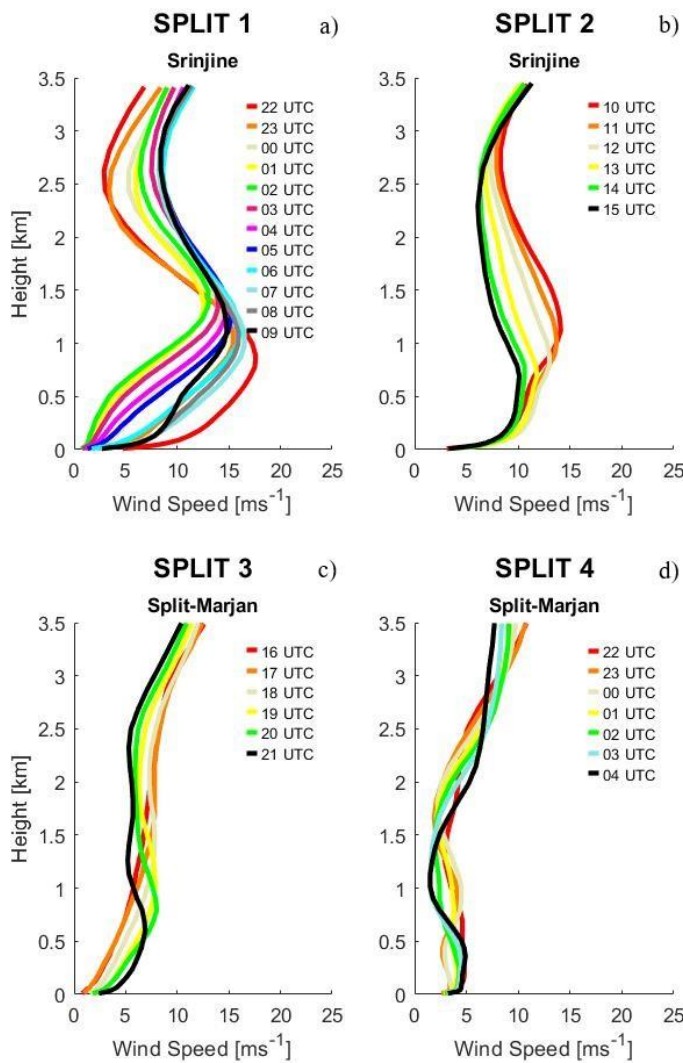

**Figure 13: Vertical profiles of wind speed (ms⁻¹) at Srinjine and Split-Marjan locations for periods (a) SPLIT 1 (from 22 UTC on 16 July 2017 to 09 UTC on 17 July 2017), (b) SPLIT 2 (from 10 UTC to 15 UTC on 17 July 2017), (c) SPLIT 3 (from 16 UTC to 21 UTC on 17 July 2017) and (d) SPLIT 4 (from 22 UTC on 17 July 2017 to 04 UTC on 18 July 2017) from ALADIN model. See Fig. 2 for location of Split-Marjan and Srinjine (noted as S).**

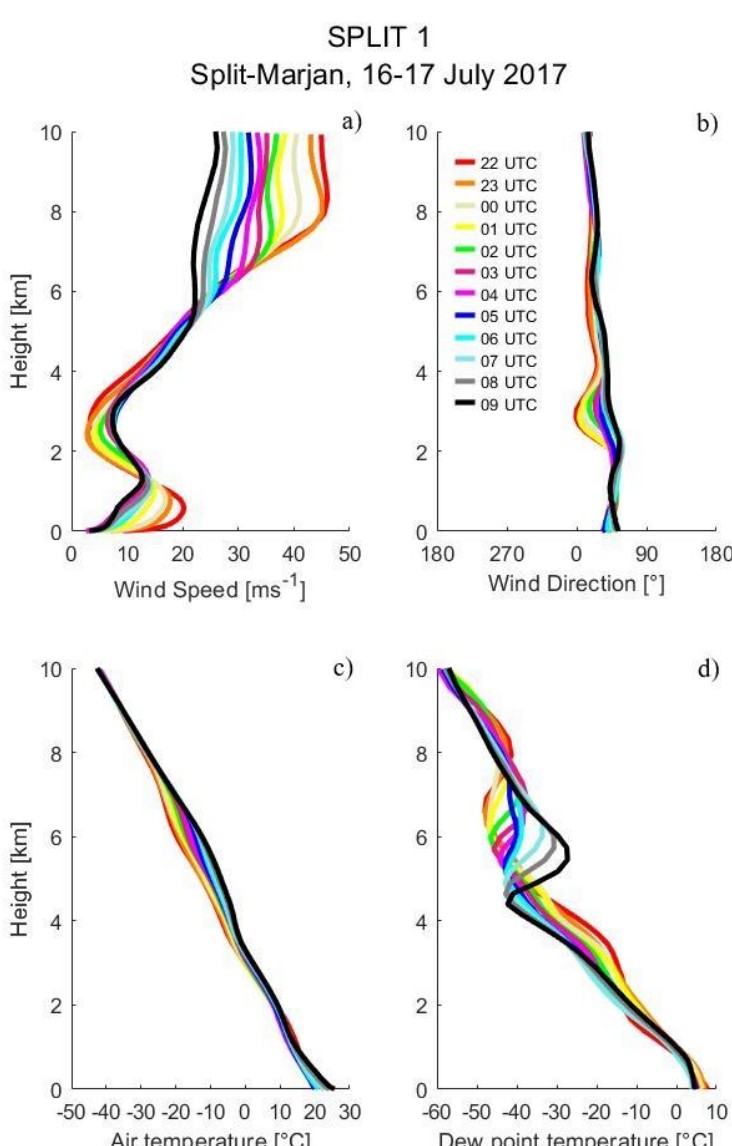

**Figure 14: Vertical profiles of (a) wind speed (ms⁻¹), (b) wind direction (°), (c) air temperature (°C) and (d) dew point temperature (°C) at Split-Marjan locations for period SPLIT 1 (from 22 UTC on 16 July 2017 to 09 UTC on 17 July 2017) from ALADIN model. See Fig. 2 for location of Split-Marjan.**


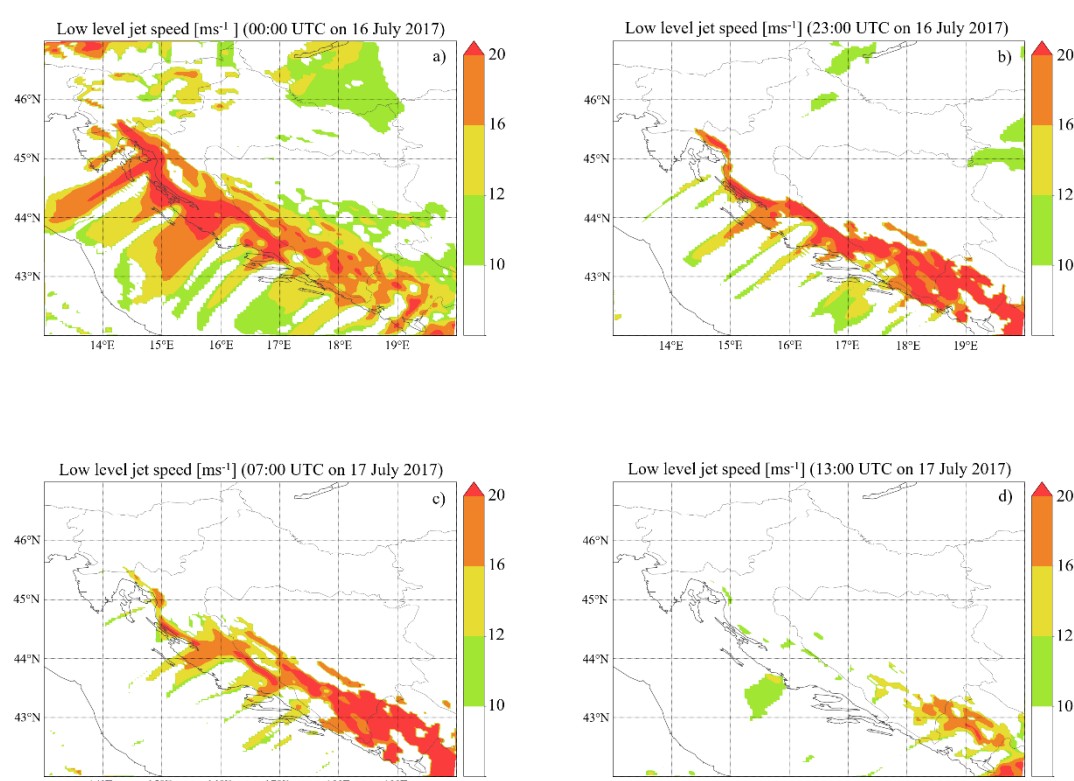


**Figure 15: Spatial distribution of low level jet defined by criterion 0 to 3 (ms⁻¹) at a) 00 UTC on 16 July 2017, b) 23 UTC on 16 July 2017 (SPLIT 1 period – ignition time), c) 07 UTC on 17 July 2017 (SPLIT 1 period – aircraft approach) and d) 13 UTC on 17 July 2017 (SPLIT 2 period) from ALADIN-HR44 model.**