# Peer review of "The 2017 Split wildfire in Croatia: Evolution and the role of meteorological conditions"

_Natural Hazards and Earth System Sciences, 2022_

## Author Response (AR1)

**Responses to the Editor**

In addition to the suggestions from the reviewers, I would like to suggest the following technical items, in most cases 'minor':

Response: We appreciate these further comments from the editor. We have also prepared responses to them, together with those for RC1 and RC2 in the following.

Abstract is so much long. Please, synthesize it. It is not necessary to go into so many details. It is enough to introduce the case study, the sources of information and methodology and the most notable results. Everything is usually written in a single paragraph in less than 20 lines.

Response: The abstract has been substantially simplified and shortened.

Line 51: add the year of the wildfires in California

Response: That was in 2018, which has been added.

Line 58. You say, "The local state of the atmosphere in days prior to wildfire and during wildfire activity is determined by synoptic scale weather systems". I am not sure that this statement is correct, as it is demonstrated throughout the article. There are several local factors that do not necessarily depend on synoptic features. For example, the slope winds to which the paper refers, or the sudden changes in wind direction. The synoptic configurations that have prevailed before the fires are very important in order to justify the state of the fuel (mainly the vegetation), but the start and spread of the fire depends mainly on mesoscale factors, as shown in the article.

Response: Thanks for pointing the inadequacy of this statement, mainly because we focused on synoptic factors in that paragraph. We have added a phrase as "The local state of the atmosphere in days prior to wildfire and during wildfire activity is determined by synoptic scale weather systems, which would determine the conditions of fuels and set the background for other mesoscale factors responsible for start and spread of fire."

Line 94. It would be better to write the sentence "Research on fire weather…" in a new paragraph. It could be the same of the July 2017 event.

Response: Revised as suggested.

Line 118. Please, clarify in the text if the number of ~2500 wildfires is the annual average (they could include very little forest fires) or the total for the period 2006-2016.

Response: It was the annual number of fires including the smallest ones. We have revised to "…in ~2500 wildfires every year (including the smallest ones) based on statistics from the period 2006-2016".

Paragraph lines 318-323. Cut-off low is usually associated to instability in high levels (and heavy rainfalls or severe weather in the eastern part, on some occasions). You speak about a "cool change", what role do you think that this cool change and this cut-off low would play?

Response: Due to the cyclonic flow of the cut-off low, the western side would have the cold and dry NNE flow as mentioned in the text. Such subsiding cold and dry air would rapidly enhance the flammability of the fuel.

Lines 338-340. "from 33.3°C to 27.0°C between 15 and 16 July. This was followed by a drop in relative humidity" Are you sure? One of the most typical situations in the great wildfires is a sudden temperature increase. A decrease of 5ºC is very strange. And a decrease of temperature would produce an increase of relative humidity for the same specific humidity, and you say that relative humidity dropped. Are you sure?

Response: This was actually the case in the observations. Bura storm over Adriatic area is characterized by a sudden decrease in air temperature and relative humidity, because cold and dry air penetrates either from the polar regions (Scandinavia) from the north of the continent or from Siberia from the northeast. Thus the strength of the storm depends on both the orography and the synoptic situation. The most frequent cases of strong bura occur in synoptic situations when there is a transitional NE state over the Adriatic area with a large pressure gradient between the rear side of the cyclone in the east and the front side of the anticyclone in the west. Therefore, the formation of a strong bura storm requires a certain synoptic situation over Europe and is not only caused by local or mesoscale processes in the atmosphere.

In the case of the Split wildfire, fresh and dry air penetrated the Adriatic area from the north, so that there was refreshment, but the air still remained dry over the Adriatic, as on the previous day, on 15 July 2017, when the maximum air temperature was 33.3 °C. Also, the vertical structure of the atmosphere using the ALADIN model showed that there was a descent of stratospheric dry air. Thus the main characteristic of a gale bura is that it is a dry and cold NE wind and this decrease in both air temperature and relative humidity is common in severe gale situations.

Lines 437-438. "Dry conditions might be explained by the complete absence of the tropopause". The tropopause exists always, although it can be upper its usual level.

Response: This is common feature known since mid-20th century and it is found in this case as well. Here are some references:

From Croatia:
Brebrić, Višnjica, 1983. Frozen rain and glaze in the region of Croatia in January 1982. Republic Hydrometeorological Service of Serbia Report, Belgrade, Yugoslavia. 289–298.

WMO Technical Note:
Berggren, R., Gibbs, W.J., & Newton, C.W., 1958. Observational characteristics of the jet stream: a survey of the literature.

NW Europe:
Vuorela, L. A., 1953. On the air flow connected with the inversion of upper tropical air over northwestern Europe, Geophysics, Helsinki, 105–130.

The tropopause drop was also evident in the case here, as simulated vertical profiles from the ALADIN model presented during 16 and most of 17 July 2017. Here we provide pseudotemp

at Split-Marjan location at 23 UTC on 16 July 2017 up to 13.5 km height (tropopause at this latitude is usually found around 10 km):

| Height (m) | Air press (hPa) | Air temp (°C) | Dew point (°C) | Wind speed (m/s) | Wind direction (°) |
|---|---|---|---|---|---|
| 13500 | 156.4 | -55.7 | -77.2 | 14.8 | 345 |
| 12904 | 171.7 | -54.1 | -73.9 | 20.5 | 353 |
| 12342 | 187.4 | -53.1 | -70.7 | 28.6 | 1 |
| 11813 | 203.5 | -52.2 | -67.8 | 35.8 | 5 |
| 11311 | 219.9 | -50.2 | -65.3 | 39.8 | 6 |
| 10831 | 236.5 | -47.4 | -63.2 | 41.8 | 6 |
| 10370 | 253.5 | -44.4 | -60.6 | 42.7 | 8 |
| 9928 | 270.7 | -41.4 | -57.1 | 43 | 10 |
| 9501 | 288.2 | -38.6 | -53 | 43 | 14 |
| 9091 | 305.9 | -36.1 | -48.9 | 43.5 | 18 |
| 8695 | 323.8 | -33.7 | -45.4 | 44.3 | 21 |
| 8312 | 341.8 | -31.1 | -43.1 | 45 | 21 |
| 7943 | 360 | -28.4 | -42.5 | 45 | 21 |
| 7585 | 378.4 | -26.2 | -43.4 | 43.4 | 21 |
| 7240 | 396.8 | -24.5 | -45.1 | 40.1 | 20 |
| 6907 | 415.4 | -23.1 | -46.8 | 36 | 19 |
| 6585 | 434 | -21.8 | -47.9 | 32.2 | 18 |
| 6275 | 452.6 | -20.4 | -47.5 | 29 | 17 |
| 5975 | 471.3 | -18.8 | -45.2 | 26.2 | 17 |
| 5685 | 490 | -17.1 | -42.3 | 23.7 | 18 |
| 5403 | 508.7 | -15.2 | -40.2 | 21.6 | 18 |
| 5131 | 527.3 | -13.3 | -38.5 | 19.8 | 20 |
| 4867 | 545.9 | -11.5 | -36.3 | 18.1 | 22 |
| 4611 | 564.3 | -9.8 | -33.6 | 16.5 | 26 |
| 4363 | 582.7 | -8.3 | -30.9 | 14.9 | 29 |
| 4124 | 601 | -6.9 | -28.1 | 13.3 | 29 |
| 3892 | 619.1 | -5.7 | -25.1 | 11.5 | 26 |
| 3668 | 637 | -4.7 | -22.2 | 9.6 | 20 |
| 3451 | 654.8 | -3.6 | -19.9 | 7.7 | 13 |
| 3242 | 672.3 | -2.1 | -18.2 | 6 | 8 |
| 3040 | 689.6 | -0.5 | -16.7 | 4.8 | 7 |
| 2844 | 706.6 | 1.1 | -15.4 | 3.8 | 9 |
| 2655 | 723.4 | 2.7 | -14.2 | 3.1 | 14 |
| 2473 | 739.9 | 4.3 | -13 | 2.8 | 24 |
| 2297 | 756 | 5.8 | -11.7 | 2.9 | 36 |
| 2127 | 771.9 | 7.3 | -10.3 | 3.6 | 45 |
| 1964 | 787.3 | 8.6 | -8.9 | 4.9 | 49 |
| 1807 | 802.4 | 9.8 | -7.6 | 6.6 | 51 |
| 1656 | 817 | 10.8 | -6.3 | 8.6 | 51 |
| 1512 | 831.3 | 11.7 | -5 | 10.5 | 51 |
| 1375 | 845.1 | 12.4 | -3.7 | 12.4 | 51 |
| 1244 | 858.4 | 13.1 | -2.4 | 14.3 | 50 |
| 1119 | 871.2 | 13.7 | -1.1 | 15.8 | 50 |

| | | | | | |
|---|---|---|---|---|---|
| 1001 | 883.5 | 14.2 | 0.2 | 16.8 | 49 |
| 889 | 895.3 | 14.7 | 1.4 | 17.5 | 48 |
| 784 | 906.5 | 15.2 | 2.6 | 17.7 | 48 |
| 685 | 917.2 | 15.7 | 3.6 | 17.7 | 46 |
| 592 | 927.2 | 16.2 | 4.4 | 17.5 | 45 |
| 506 | 936.6 | 16.7 | 5 | 17.2 | 44 |
| 426 | 945.4 | 17.2 | 5.5 | 16.8 | 43 |
| 353 | 953.6 | 17.6 | 5.9 | 16.3 | 41 |
| 287 | 961 | 18.1 | 6.2 | 15.8 | 40 |
| 227 | 967.7 | 18.5 | 6.4 | 15.1 | 39 |
| 174 | 973.7 | 18.8 | 6.6 | 14.4 | 39 |
| 128 | 979 | 19.2 | 6.7 | 13.5 | 39 |
| 89 | 983.5 | 19.4 | 6.8 | 12.4 | 39 |
| 62 | 986.6 | 19.6 | 6.9 | 11.3 | 39 |
| 43 | 988.8 | 19.7 | 6.9 | 10.2 | 39 |
| 26 | 990.7 | 19.8 | 7 | 8.7 | 40 |
| 9 | 992.6 | 19.9 | 7 | 7 | 42 |

Referee 1 says": Perhaps consider summarizing in the conclusions section the uniqueness of the environmental conditions enabling and driving this fire." I agree with him/her and reading your answer I am afraid that you have not understood him/her. Concluding remarks are so much long. I would propose you to focus in the meteorological and climatic (for antecedent conditions) factors that leaded to these great wildfires in Croatia in 2017, avoiding references and discussion. Discussion aspects, including comparison with other situations, could be moved to subsections 5.1. and 5.2

Response: Thank you for this clarification about our understanding of RC1's comment. Accordingly, we have prepared a concise summary of the unique drivers of the event at the beginning of section 5.3, and the discussion in that section has also been simplified (e.g., discussion on LLJ has been merged to 5.2 in response to RC2).

**Responses to RC1:**

The paper is very well written and could be accepted as is. It is good to see this type of research for Croatia. Below are two very minor comments for consideration:

Response: We appreciate the appraisal from the reviewer, and the support for this study for Croatia that has not been exposed much in the literature. When we prepare the final manuscript we will revise according to these minor comments as in our responses below.

Line 53: Perhaps indicate if this is a regional or otherwise statistic so that readers do not necessarily have to look up the reference.

Response: This statement was based on Strauss et al. (1989), which has validated some theoretical fire size distributions by data from western U.S., and thus regional statistics. The statement has been revised to "Extreme wildfires are rare, based on statistics from western U.S. they account for only 1 % of fire occurrences but cause more than 90% of damage (Strauss et al., 1989)."

Line 60: Perhaps consider adding a more recent critical fire weather pattern publication such as https://landscapepartnership.org/maps-data/climate-context/cc-resources/ClimateSciPDFs/JFSP%20Extreme%20Fire.pdf/app-download-file/file/JFSP%20Extreme%20Fire.pdf#page=37

Response: Thank you for the suggested additional reference. We have quoted this reference (Werth 2011) in the statement.

General: Perhaps consider summarizing in the conclusions section the uniqueness of the environmental conditions enabling and driving this fire.

Response: Thanks for this suggestion. This section 5 has not been organized well enough before. The unique environmental conditions enabling and driving this fire have been detailed in 5.1. To provide a concise summary of the unique environmental conditions and drivers of this event, we have added several dot points at the beginning of 5.3. The discussion in the same section has also been simplified according to the comment from the editor.

**Response to RC2:**

General comments
The comprehensive analysis and diagnosis of weather and associated fire behaviour of the 2017 Split wildfire in Croatia were presented in this study. The manuscript is well written, and the analysis and discussions flow well. The scientific questions are well defined in the study and the results from analysis presented as figures and tables provide good supports to the answers.

Response: We appreciate the appraisal from the reviewer. In the final manuscript we have revised according to your minor comments as in our responses below.

Very happy to see the robust discussions of how synoptic, mesoscale, and local weather influence fire behaviour in this study. But we know wildfires would also simultaneously impact weather, hope to see there's a future study on the feedback from Split wildfire to the local weather too.

Response: Thank you for this insightful comment. We agree that the heat and moisture from the wildfire event may has modified the weather (which consisted of the large pressure gradient and the LLJ). We are performing numerical modelling of the event using both weather model and weather-fire coupled model, and based on which will diagnose such feedback processes. In the second to last paragraph of section 5.3 (concluding remarks), we have briefly discussed this and in the revised manuscript we have further elaborated:

"In recent decades extreme wildfires around the world have demonstrated their destructive power, creating even their own weather and producing dangerous phenomena such as fire whirls, tornadoes or fire storms generated from pyroconvection (Tory et al., 2018; Tory and Kepert 2021) generally. As the Split wildfire also demonstrated unprecedented fire behavior, it is very likely that the energy released from the wildfire influenced the meteorology and surrounding atmosphere (e.g., Peace et al. 2015). To investigate this matter, it is in addition plan to prepare coupled fire-atmosphere simulations for this case study."

A Few technical comments are listed below
1. Figure 8 and Figure 10 are not referenced in the main text

   Response: Thanks for pointing this out. We have added quotation of Fig. 8a in the first paragraph of section 4.5 (Figs. 8b, c have been quoted in third paragraph). Quotation to Fig. 10 has been added in the fourth paragraph.

2. Lines 530 to 553 seems only presenting discussions on Bura (section 5.2), and discussion on LLJ appear after line 554 (section 5.3). If that is the case, title of section should be amended to only include Bura.

   Response: Thanks for this suggestion and we are sorry that section 5 has not been organized very well. As you mentioned, there was discussion on LLJ after 5.2, and thus we have put the several paragraphs back to 5.2.
   Also based on a comment from RC1 and the editor, we have prepared a concise summary of the unique driver of this event at the beginning of 5.3 Concluding remarks. That subsection has also been simplified.